# Zebrafish *tsc1* and *cxcl12a* increase susceptibility to mycobacterial infection

Kathryn Wright[1,2,3,4], Darryl JY Han[5], Renhua Song[4,6], Kumudika de Silva[2], Karren M Plain[2], Auriol C Purdie[2] , Ava Shepherd[3] , Maegan Chin[3], Elinor Hortle[1,4,7] , Justin J-L Wong[4,6], Warwick J Britton[1,4,8] , Stefan H Oehlers[1,4,5] 

Regulation of host miRNA expression is a contested node that controls the host immune response to mycobacterial infection. The host must counter subversive efforts of pathogenic mycobacteria to launch a protective immune response. Here, we examine the role of miR-126 in the zebrafish–*Mycobacterium marinum* infection model and identify a protective role for infection-induced miR-126 through multiple effector pathways. We identified a putative link between miR-126 and the *tsc1a* and *cxcl12a/ccl2/ccr2* signalling axes resulting in the suppression of non-*tnfa* expressing macrophage accumulation at early *M. marinum* granulomas. Mechanistically, we found a detrimental effect of *tsc1a* expression that renders zebrafish embryos susceptible to higher bacterial burden and increased cell death via mTOR inhibition. We found that macrophage recruitment driven by the *cxcl12a/ccl2/ccr2* signalling axis was at the expense of the recruitment of classically activated *tnfa*-expressing macrophages and increased cell death around granulomas. Together, our results delineate putative pathways by which infection-induced miR-126 may shape an effective immune response to *M. marinum* infection in zebrafish embryos.

## Introduction

Manipulation and subversion of host signalling pathways is a hallmark of infection by pathogenic mycobacteria, such as the causative agents of tuberculosis and leprosy (Hortle & Oehlers, 2020; Ramakrishnan, 2020). Infection with pathogenic mycobacteria results in dysregulated immune responses and disruption of key signalling cascades, ultimately supporting the survival and persistence of bacteria (Stutz et al, 2018). A crucial element of mycobacterial persistence is the interaction between bacteria and innate immune cells, primarily macrophages.

Subversion of this normally protective interaction between macrophage and bacterium leads to the formation of granulomas which aid bacterial survival. Pleiotropic modulation of host gene expression and signalling cascades by miRNA potentially act as fine-tuners of the immune cell response to mycobacteria and may shape the outcome of infection.

miRNAs are short single-stranded non-coding RNA molecules that post-transcriptionally regulate gene expression. Through a process known as gene-silencing, miRNAs bind to the UTRs of target mRNA and reduce their stability. Depending on the degree of complementarity in base pairing, miRNA may either degrade the bound mRNA or transiently suppress translation, reducing protein production (Duchaine & Fabian, 2019). As fine-tuners of gene expression, miRNAs have been investigated in multiple pathologies to identify their functional pathways and biomarker potential (Correia et al, 2017; Wang et al, 2018). During mycobacterial infections, miRNAs are differentially regulated and play significant biological roles during infection (Wright et al, 2019, 2021; Han et al, 2020; Ruiz-Tagle et al, 2020; Mehta, 2021). The regulation of miRNA expression is a contested node, in that it is multifaceted and driven by several opposing factors. Host control of miRNA is challenged by pathogen-driven modulation of expression, and the downstream impacts on host immunity may result in either successful clearance or subversion by the pathogen to support infection.

Reduced expression of miR-126 has been reported in cattle suffering from Johne's disease, an infection caused by *Mycobacterium avium* subspecies *paratuberculosis*; in *Mycobacterium abscessus* and *Mycobacterium tuberculosis*–infected THP-1 cells; and in patients with tuberculous meningitis, a severe manifestation of *M. tuberculosis* infection, and pulmonary tuberculosis (Aulicino et al, 2015; Barry et al, 2018; Gupta et al, 2018; Pan et al, 2019; Wang et al, 2020). Although mapped targets suggest further roles in haematopoiesis and inflammatory disorders, the function of miR-126 in infection has yet to be defined (Shen et al, 2008; Zernecke et al, 2009; Oglesby et al, 2010; Grabher et al, 2011).

[1]Tuberculosis Research Program at the Centenary Institute, The University of Sydney, Camperdown, Australia   [2]Faculty of Science, Sydney School of Veterinary Science, The University of Sydney, Sydney, Australia   [3]Directed Evolution Research Program at the Centenary Institute, The University of Sydney, Camperdown, Australia   [4]Faculty of Medicine and Health, The University of Sydney, Camperdown, Australia   [5]A*STAR Infectious Diseases Labs (A*STAR ID Labs), Agency for Science, Technology and Research (A*STAR), Singapore, Singapore   [6]Epigenetics and RNA Biology Laboratory, Charles Perkins Centre, The University of Sydney, Camperdown, Australia   [7]Faculty of Science, School of Life Sciences, Centre for Inflammation and University of Technology Sydney, Sydney, Australia   [8]Department of Clinical Immunology, Royal Prince Alfred Hospital, Camperdown, Australia

Correspondence: stefan_oehlers@idlabs.a-star.edu.sg

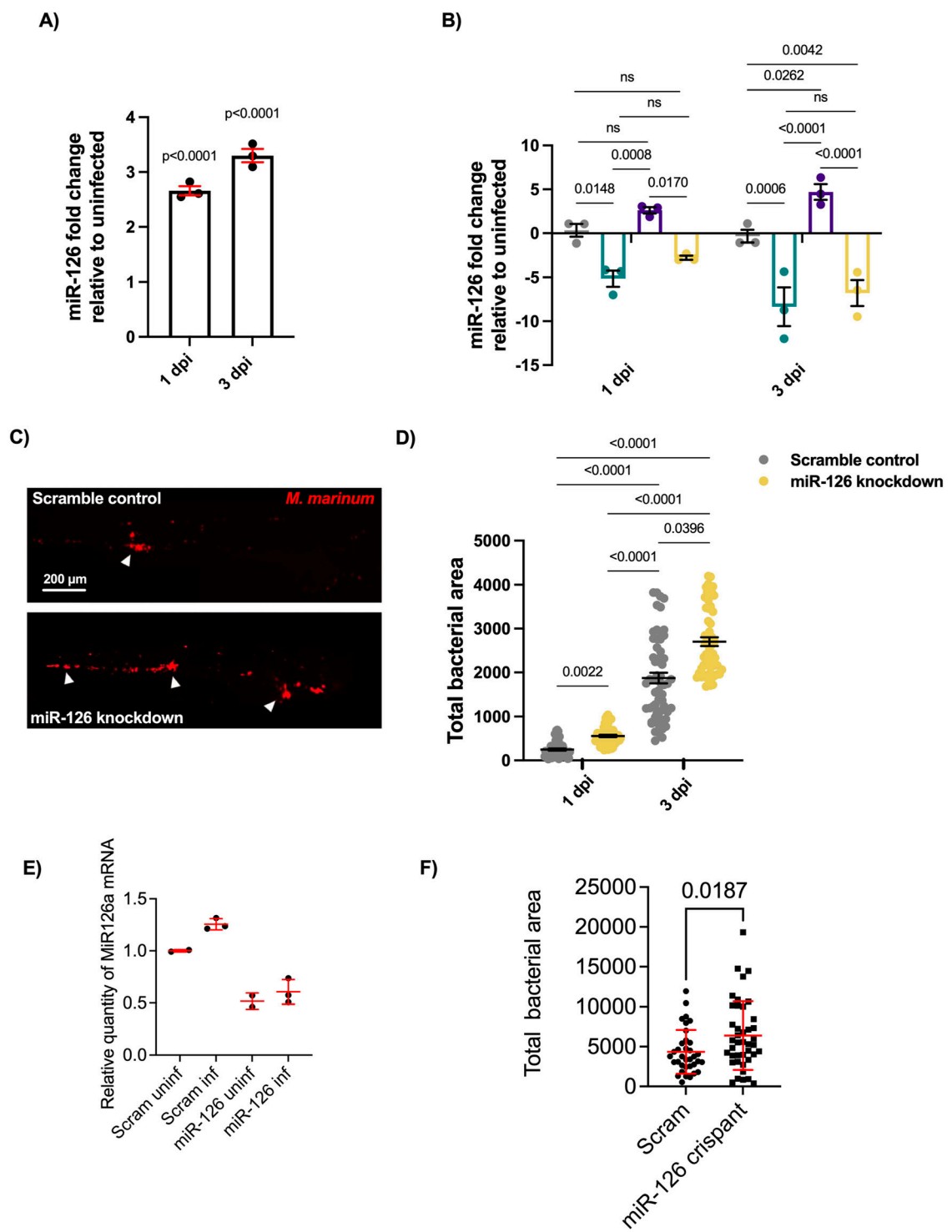

**Figure 1. Infection-induced miR-126 expression alters bacterial burden.**
**(A)** Expression of miR-126 after *M. marinum* infection analysed by RT-qPCR at 1 and 3 dpi relative to uninfected embryos. **(B)** Expression of miR-126 in uninfected and infected, antagomir-injected (miR-126 knockdown), and scramble-injected embryos. **(C)** Representative images of *M. marinum* infection at 3 dpi in scramble control and miR-126 knockdown embryos. Scale bar represents 200 μm. **(D)** *M. marinum* burden in miR-126 knockdown embryos at 1 and 3 dpi. **(E)** Expression of miR-126 in uninfected and infected gRNA/Cas9-injected embryos at 5 dpi. **(F)** *M. marinum* burden in miR-126 crispant embryos at 5 dpi. Data information: each data point represents a single measurement, with the mean and SEM shown. For RT-qPCR analysis, each data point represents 10 embryos and contains three biological replicates. Bacterial burden analysis data points represent individual embryos (n = 40–50 embryos per group) and are representative of two experimental replicates.

Zebrafish (*Danio rerio*) is a powerful model organism for studying host–mycobacterial interactions as they allow direct visualisation of cellular interactions and are amenable to straightforward genetic manipulation. An added benefit of the zebrafish–*Mycobacterium marinum* model is the ability to use a native pathogen of the zebrafish, which closely mirrors histological aspects of human–*M. tuberculosis* pathogenesis (Davis et al, 2002; Stinear et al, 2008). Zebrafish provide an established model for the investigation of miRNA function and responses to infection, with well-developed genetic tools that facilitate the study of downstream host gene function. We have recently used the zebrafish–*M. marinum* model to investigate the downstream targets of miR-206 and identified the role of this conserved miR in controlling the multicellular immune response to acute disseminated mycobacterial infection (Wright et al, 2021).

Here, we use the zebrafish–*M. marinum* platform to examine the role of mycobacterium infection-induced dre-miR-126a-3p (miR-126). We link infection-induced miR-126 to Tsc1 via activation of the mTOR pathway and improved macrophage function because of regulation of a Cxcl12/Ccl2/Ccr2 signalling axis during the early stages of mycobacterial infection.

# Results

## Infection-induced miR-126 is host-protective against *M. marinum* infection

Embryos were infected with *M. marinum* via caudal vein injection and analysed at 1 and 3 days post infection (dpi) by quantitative (q) PCR to assess the responsiveness of miR-126 expression to infection. At both timepoints, miR-126 was up-regulated in *M. marinum*–infected embryos in comparison to uninfected control embryos (Fig 1A).

To determine if antagomiR-mediated knockdown of miR-126 effectively reduced transcript abundance and if this miR-126 knockdown was sustained after *M. marinum* infection, embryos were injected with antagomiR at the single-cell stage and infected with *M. marinum* at 1.5 days post fertilisation (dpf). Expression levels of miR-126 were measured at 1 and 3 dpi. AntagomiR knockdown reduced miR-126 levels at both timepoints and prevented the infection-associated miR-126 expression (Fig 1B).

The effect of miR-126 expression on mycobacterial infection was assessed through analysis of bacterial burden after infection in control and miR-126 knockdown embryos (Fig 1C). There was a small, but statistically significant, increase in the bacterial burden of miR-126 knockdown embryos at 1 dpi, which was much more apparent at 3 dpi (Fig 1D). We next used CRISPR-Cas9 to validate the effects of antagomir injection. Injection with guide RNAs targeting the miR-126 locus and Cas9 reduced the amount of miR-126 detectable by RT–qPCR indicating a possible combination of primer site mutation and transcript decay compared with embryos injected with scrambled guide RNAs and Cas9 (Fig 1E). Furthermore, miR-126 crispants had a significantly increased bacterial burden compared with scramble control embryos (Fig 1F). These results indicate that *M. marinum* infection induces up-regulation of miR-126 and that miR-126 has a host-protective effect.

The bacterial burden measurement uses the transparent nature of larval zebrafish and allows for live imaging of infected embryos. To ensure measuring fluorescent pixels provided an accurate view of infectious burden, the fluorescent bacterial area was compared with conventional CFU recovery assay. At both 1 and 3 dpi, bacterial burden measurements by fluorescent bacterial area displayed the same trends as CFU enumeration and provided a comparable, immediate measurement. This measurement allows for continued assaying of the same embryo, and hence, bacterial burden measurement by the fluorescent bacterial area was used for all larval infection experiments (Fig S1A and B).

## miR-126 target gene mRNA expression patterns are conserved during *M. marinum* infection of zebrafish

To uncover the biological pathways targeted by miR-126, expression of potential target mRNAs was analysed by RT-qPCR after antagomiR knockdown and infection with *M. marinum*. Possible target genes were chosen from published experimentally observed targets and bioinformatic target prediction software (Supplemental Data 1), based on their proven involvement in immune responses to bacterial infections and roles in key immune cell signalling (Grimson et al, 2007; Fish et al, 2008; Meng et al, 2012; Ulitsky et al, 2012; Li et al, 2013; Zhang et al, 2013; Agudo et al, 2014; Liu et al, 2014a, 2014b; Yuan et al, 2018; Huang et al, 2020). Increased expression of a transcript in miR-126 knockdown embryos compared with uninfected scrambled controls was expected to indicate targeting by miR-126 (Fig 2A–D).

Expression of *cxcr4b* was significantly increased in *M. marinum*–infected and miR-126 knockdown + *M. marinum* embryos compared with control at 1 dpi (Fig 2E–H). Expression of *cxcr4a* was increased in all treatments but was not significantly different from control uninfected embryos. Expression of *cxcl12a* was higher in both miR-126 knockdown and *M. marinum*–infected embryos than the control but not knockdown infected embryos. miR-126 knockdown + *M. marinum* embryos had reduced *cxcl12a* expression compared with control infected embryos. Although expression of *spred1* was increased in both knockdown groups compared with the control, there was no difference between knockdown infected and control infected embryos. Finally, *tsc1a* was increased in all treatments from control embryos, although expression was significantly higher compared with *M. marinum* alone in miR-126 knockdown + *M. marinum* at 1 dpi (Fig 2D) and both mir-126 knockdown and miR-126 knockdown + *M. marinum* at 3 dpi (Fig 2H).

From the gene expression analysis, *cxcr4b*, *cxcl12a*, and *tsc1a* were regarded as potential target genes of miR-126 because of their increased expression in knockdown and knockdown infected embryos and differential regulation compared with *M. marinum*–infected embryos. Despite no increase in expression at 1 dpi of *cxcr4b* in miR-126 knockdown alone, the increased expression in miR-126 knockdown infected embryos made *cxcr4b* a gene of interest alongside *cxcl12a* because of their previously identified role in mycobacterial and zebrafish immunity (Torraca et al, 2017b; Isles et al, 2019; Wright et al, 2021). However, the late increase in *cxcr4b* expression at 3 dpi suggests this gene may not be a direct target of miR-126 but rather a further downstream molecule that may yet still represent a significant molecule in miR-126–dependant signalling

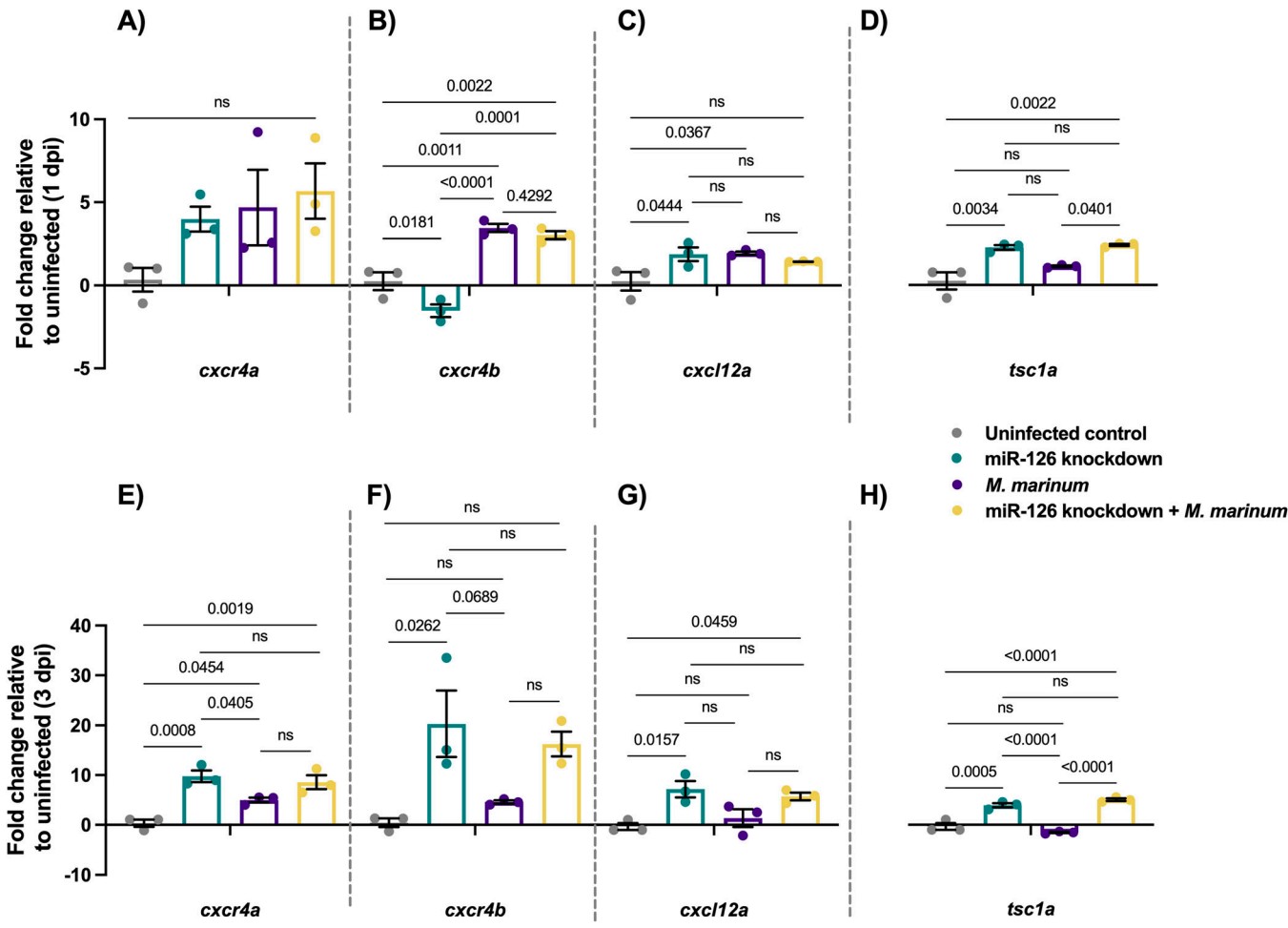

**Figure 2. Expression of potential miR-126 mRNA targets is conserved in zebrafish *M. marinum* infection.**
**(A)** Expression of *cxcr4a* at 1 dpi in miR-126 knockdown measured by RT-qPCR relative to uninfected scramble control embryos. **(B)** Expression of *cxcr4b* at 1 dpi in miR-126 knockdown measured by RT-qPCR relative to uninfected scramble control embryos. **(C)** Expression of *cxcl12a* at 1 dpi in miR-126 knockdown measured by RT-qPCR relative to uninfected scramble control embryos. **(D)** Expression of *tsc1a* at 1 dpi in miR-126 knockdown measured by RT-qPCR relative to uninfected scramble control embryos. **(E)** Expression of *cxcr4a* at 1 dpi in miR-126 knockdown measured by RT-qPCR relative to uninfected scramble control embryos. **(F)** Expression of *cxcr4b* at 1 dpi in miR-126 knockdown measured by RT-qPCR relative to uninfected scramble control embryos. **(G)** Expression of *cxcl12a* at 1 dpi in miR-126 knockdown measured by RT-qPCR relative to uninfected scramble control embryos. **(H)** Expression of *tsc1a* at 1 dpi in miR-126 knockdown measured by RT-qPCR relative to uninfected scramble control embryos. Data information: each data point represents a single measurement of 10 pooled embryos and two technical replicates, with the mean and SEM shown and comparisons calculated by one-way ANOVA.

pathways. Also of interest, *tsc1a* may be involved in mycobacterial pathogenesis as a negative regulator of the mTOR signalling pathway, a key regulatory pathway of a variety of cellular functions (Cardenal-Muñoz et al, 2017; Pagan et al, 2022). Expression of genes of interest *cxcr4a*, *cxcr4b*, *cxcl12a*, and *tsc1a* was measured at 3 dpi and showed that knockdown of miR-126 increased the expression of all target genes even in *M. marinum*–infected knockdown embryos, when compared with control uninfected and *M. marinum*–infected only embryos (Fig 2E–H).

RNA sequencing analysis of total RNA from miR-126 antagomir and crispant embryos yielded a strong neurobiology and notch signalling signature as expected from the literature (Fig S2A–2D) (Shen et al, 2008; Zernecke et al, 2009; Oglesby et al, 2010; Grabher et al, 2011). However, RNA sequencing did not detect differential regulation of the immune genes predicted in silico or observed by

RT–qPCR even when carried out on the same total RNA samples (Supplemental Data 1). Importantly, this precludes inference of direct interaction between miR-126 and the predicted immune targets in vivo.

### Suppression of *tsc1a* expression aids control of *M. marinum* infection

After the RT–qPCR expression profiling data, *tsc1a* was selected for further investigation because of differential expression between *M. marinum*–infected and miR-126 knockdown embryos. Predicted binding of miR-126 and *tsc1a* in zebrafish is summarised in (Fig 3A).

To investigate the role of *tsc1a* in infection, we targeted *tsc1a* for knockdown using CRISPR-Cas9 (Fig 3B). Knockdown of *tsc1a* significantly reduced transcript abundance at 1 and 3 dpi compared

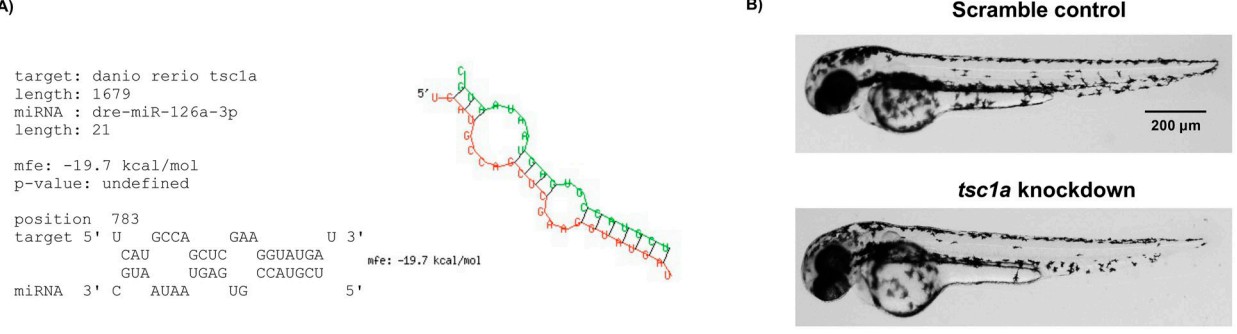

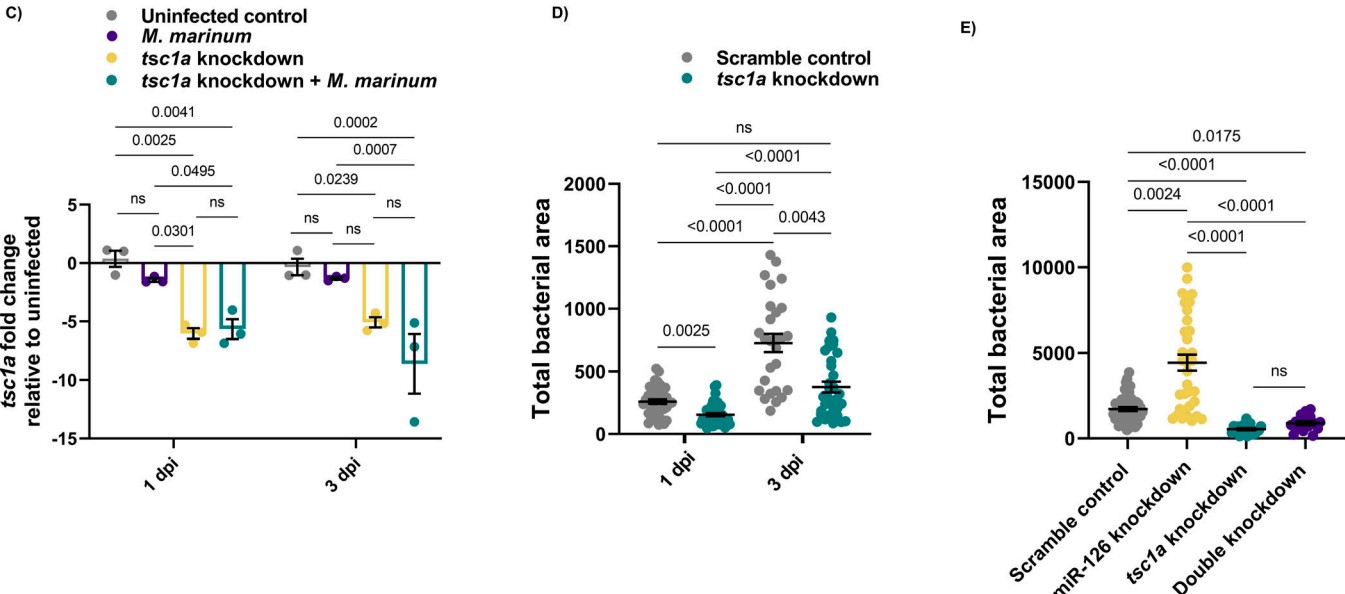

**Figure 3. miR-126 potentially targets *tsc1a* to worsen *M. marinum* infection burden.**
**(A)** Binding kinetics of *tsc1a* and dre-miR-126a-3p as predicted by RNAhybrid. **(B)** Brightfield images of *tsc1a* knockdown embryos at 3 dpf showing no abnormal developmental phenotypes. Scale bar represents 200 μm. **(C)** *tsc1a* expression was measured by RT-qPCR at 1 and 3 dpi after CRISPR-Cas9 knockdown of *tsc1a* and infection with *M. marinum*. **(D)** *tsc1a* knockdown and scramble control embryos were infected with *M. marinum* via caudal vein injection, and the bacteria burden was analysed at 1 and 3 dpi. **(E)** *tsc1a* and miR-126 double knockdown embryos were infected with *M. marinum* via caudal vein injection, and bacterial burden was analysed at 3 dpi. Data information: for RT-qPCR analysis, data points are representative of a single measurement of 10 pooled embryos and two technical replicates, with the mean and SEM shown and comparisons calculated by one-way ANOVA. **(C, D)** Bacterial burden data points represent a single measurement (n = 27–44 embryos per group (C) and n = 11–20 embryos per group (D)), and two experimental replicates with the mean and SEM are shown and comparisons calculated by one-way ANOVA.

with both control uninfected and *M. marinum*–infected embryos (Fig 3C). At both timepoints, *tsc1a* knockdown was sustained in *M. marinum*–infected knockdown embryos. Knockdown of *tsc1a* significantly reduced *M. marinum* burden compared with control embryos (Fig 3D). Double knockdown of both miR-126 and *tsc1a* significantly reduced bacterial burden compared with miR-126 knockdown alone. There was no difference in bacterial burden between double knockdown embryos and *tsc1a* knockdown alone embryos, suggesting that *tsc1a* is driving the miR-126 knockdown-associated increase in burden (Fig 3E). The opposing effects of miR-126 knockdown and *tsc1a* knockdown on *M. marinum* burden suggested involvement of a potential miR-126/Tsc1 signalling axis in mycobacterial infection.

### Knockdown of *tsc1a* alters downstream host immune pathways and increases host defence during infection

To further understand the impact of altered *tsc1a* levels during infection and the impact on downstream targets, RNA sequencing was performed on *tsc1a* knockdown and *M. marinum*–infected *tsc1a* knockdown infected embryos (Supplemental Data 2). Knockdown of *tsc1a* significantly altered the transcriptomic landscape, leading to widespread changes in transcript abundance (Fig 4A). To further identify key pathways, a gene set enrichment analysis was performed, revealing a strong enrichment for the mTOR signalling pathway in *tsc1a* knockdown embryos, substantiating the role of Tsc1 as a negative regulator of mTOR (Fig 4B). The observed

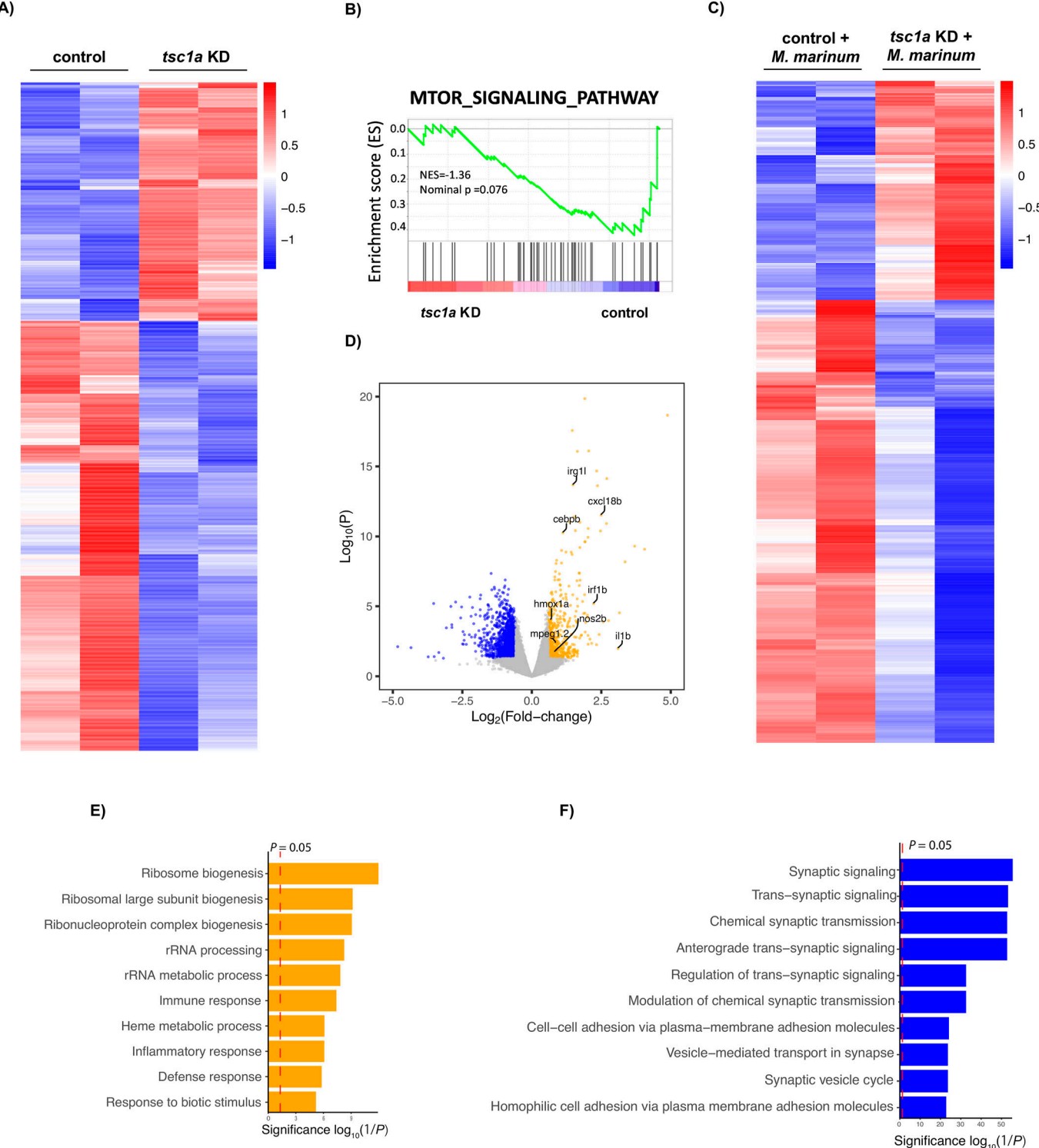

**Figure 4.  Modulation of Tsc1 alters key host defence and inflammatory pathways.**
**(A)** Heatmap of differentially expressed genes (DEGs) identified by RNA sequencing of 3 dpf embryos after CRISPR-Cas9 knockdown of *tsc1a*. **(B)** Enrichment score mapping to the term MTOR signalling pathway of DEG identified by RNA sequencing of 3 dpf embryos after CRISPR-Cas9 knockdown of *tsc1a*. **(C)** Heatmap of DEGs identified by RNA sequencing of 3 dpi embryos after CRISPR-Cas9 knockdown of *tsc1a* and infection with *M. marinum*. **(D)** Volcano plot of DEGs identified by RNA sequencing of 3 dpi embryos after CRISPR-Cas9 knockdown of *tsc1a* and infection with *M. marinum*. Up-regulated host defence and inflammatory pathway genes are annotated. **(E, F)** Gene ontology analysis was performed on DEGs identified by RNA sequencing of 3 dpi embryos after CRISPR-Cas9 knockdown of *tsc1a* and infection with *M. marinum* to identify key downstream pathways either increased (E) or decreased (F) in *tsc1a* crispants.

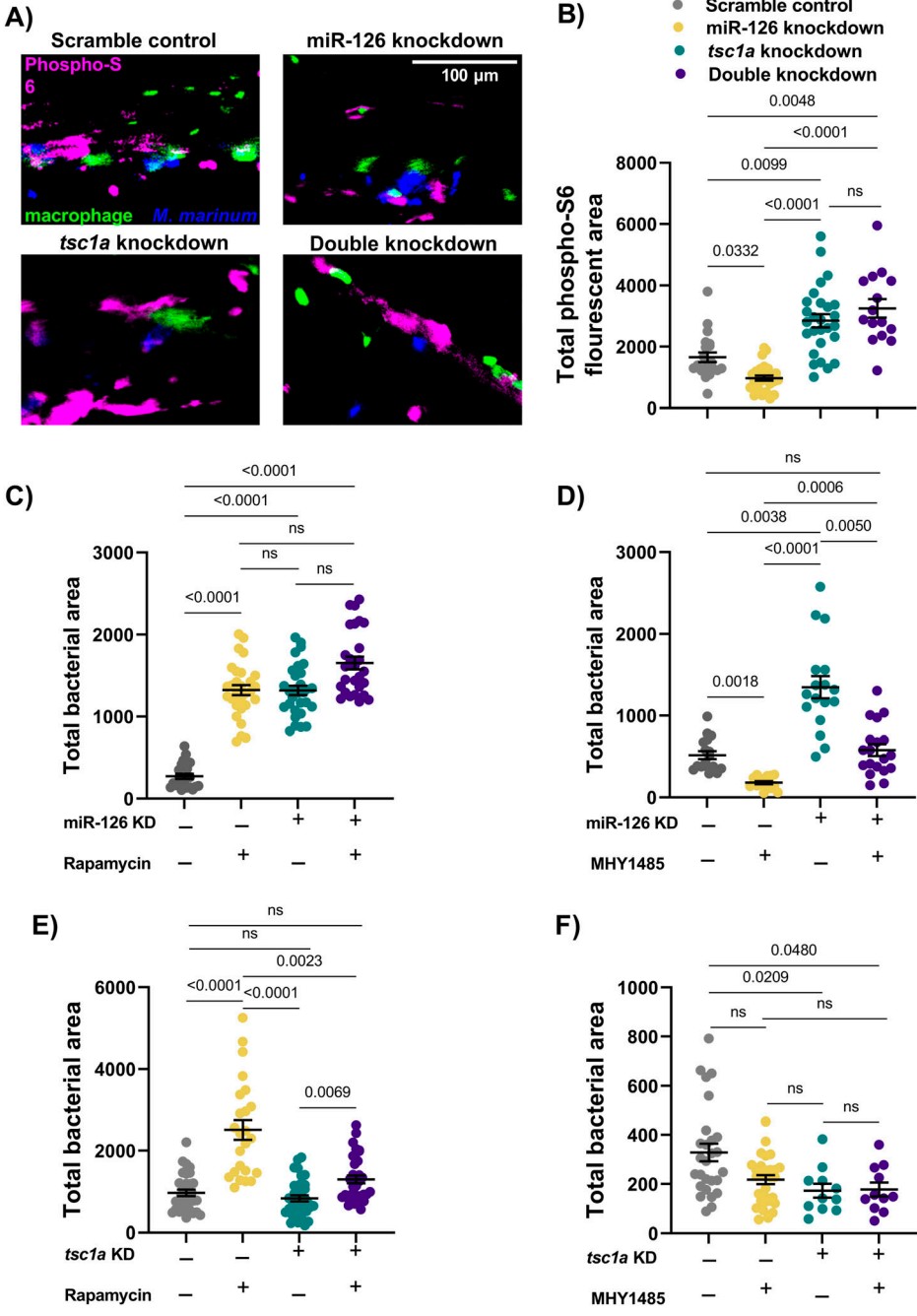

**Figure 5.** miR-126 acts on *tsc1a* to influence mTOR signalling during infection.

**(A)** Representative images of phospho-S6 fluorescent staining in miR-126 and *tsc1a* knockdown embryos at 1 dpi. *M. marinum* is blue, macrophages are green, and phosphorylated ribosomal protein S6 is magenta. Scale bar represents 100 *μm*. **(B)** Phospho-S6 staining in *M. marinum*–infected scramble control, miR-126 knockdown, and *tsc1a* knockdown at 1 dpi. **(C)** miR-126 knockdown embryos were infected with *M. marinum* via caudal vein injection and treated with mTOR inhibitor rapamycin. Bacterial burden was analysed at 1 dpi. **(D)** miR-126 knockdown embryos were infected with *M. marinum* via caudal vein injection and treated with mTOR activator MHY1485. **(E)** *tsc1a* knockdown embryos were infected with *M. marinum* via caudal vein injection and treated with mTOR inhibitor rapamycin. Bacterial burden was analysed at 1 dpi. **(F)** *tsc1a* knockdown embryos were infected with *M. marinum* via caudal vein injection and treated with mTOR activator MHY1485. Bacterial burden was analysed at 1 dpi. Data information: each data point represents a single measurement (n = 11–36 embryos per group) with the mean and SEM shown. Phospho-S6 staining is a single experimental replicate, whereas rapamycin and MHY1485 treatments represent two experimental replicates each.

transcriptional changes observed in *tsc1a* embryos are sustained during infection with *M. marinum* (Fig 4C). Increased expression of key inflammatory and bactericidal genes including *irg1l*, *cxcl18b*, and *il1b* was also observed, suggesting an increased inflammatory response to *M. marinum* infection in *tsc1a* knockdown embryos (Fig 4D) (Hall et al, 2013; Jayaraman et al, 2013; Torraca et al, 2017a). Gene ontology analysis further revealed an increase in pathways associated with mTOR signalling including ribosome biogenesis and rRNA processes and increased gene expression associated with inflammatory responses and host defence (Fig 4E) (Iadevaia et al, 2012, 2014; Weichhart et al, 2015). As expected in *tsc1a* knockdown

embryos, decreased expression of genes correlating with synaptic activity was observed, a common phenotype in the condition tuberous sclerosis, caused by mutations and disruption of Tsc transcripts (Fig 4F) (Litwa, 2022).

### Knockdown of miR-126 or *tsc1a* alters the mTOR signalling axis in *M. marinum* infection

As *tsc1a* encodes a negative regulator of mTOR function, we investigated downstream mTOR activity in miR-126 and *tsc1a* knockdown embryos. We used whole-mount embryo immunofluorescent staining

of phosphorylated ribosomal protein S6 (phospho-S6) as a readout for mTOR activity after infection of miR-126 and *tsc1a* knockdown embryos with *M. marinum* (Fig 5A). At 1 dpi, knockdown of miR-126 resulted in reduced phospho-S6 staining, consistent with increased *tsc1a* transcript abundance and decreased mTOR function, and knockdown of either *tsc1a* alone or double knockdown of miR-126 and *tsc1a* increased phospho-S6 staining compared with both control and miR-126 knockdown embryos placing *tsc1a*-mediated mTOR activation downstream of miR-126 (Fig 5B). Although phospho-S6 staining was present around fluorescent macrophages, the bulk changes in phospho-S6 staining occurred largely distal to the fluorescent *M. marinum*, with no observed colocalisation, suggesting a lack of specificity to infection (Fig 5A).

To assess the impact of decreased mTOR signalling in miR-126 knockdown, embryos were treated with either an inhibitor (rapamycin) or activator (MHY1485) of mTOR. Treatment of control embryos with rapamycin resulted in a similar increased burden to that seen after knockdown of miR-126 (Fig 5C). The combination of miR-126 knockdown and rapamycin treatment did not significantly alter the bacterial burden from miR-126 knockdown alone. Although there appeared to be a non-significant trend towards increasing bacterial burden in rapamycin-treated miR-126 knockdown embryos at 1 dpi, this was not observed by 3 dpi, which we attribute to the strong and immediate immunosuppressive effects of rapamycin.

Conversely, activation of mTOR signalling with MHY1485 significantly reduced the bacterial burden in both control and miR-126 knockdown backgrounds (Fig 5D). Treatment of miR-126 knockdown embryos with MHY1485 only partially rescued the miR-126 knockdown-induced increase in bacterial burden as compared with MHY1485 treatment alone, indicating that these pathways are not fully interlinked and that multiple miR-126–dependant mechanisms are functioning during infection.

As anticipated, inhibition of mTOR signalling through rapamycin treatment of *tsc1a* knockdown embryos increased bacterial burden, however, not to the levels seen in rapamycin-treated scramble control embryos (Fig 5E). Also as anticipated, treatment with MHY1485 did not further decrease burden levels from *tsc1a* knockdown alone (Fig 5F). Together, these data place mTOR activation downstream of a putative infection-induced miR-126/*tsc1a* pathway and demonstrate that additional non-mTOR factors mediate the infection phenotype downstream of the miR-126/*tsc1a* axis.

### miR-126 knockdown and inhibition of mTOR increase cell death in *M. marinum* infection

Because of the pleiotropic effects of mTOR on apoptosis and cell death, we hypothesised that decreased mTOR activity caused by increased *tsc1a* expression in miR-126 knockdown embryos might compromise the survival of *M. marinum*–infected macrophages (Byles et al, 2013; Dossou & Basu, 2019; Zhang et al, 2019; Pagan et al, 2022). To explore this hypothesis, we performed TUNEL staining on 3 dpi miR-126 knockdown embryos (Fig 6A). Knockdown of miR-126 did increase the number of TUNEL-stained cells compared with control infected embryos (Fig 6B).

The increased cell death in miR-126 knockdown embryos was completely reversed by double knockdown of miR-126 and *tsc1a* indicating that the cell death phenotype is dependent on *tsc1* expression (Fig 6C and D). Rapamycin treatment increased the number of TUNEL-stained cells in control and *tsc1a* knockdown embryos; however, there was no significant increase in the miR-126 knockdown embryos as expected from the bacterial burden data (Fig 6E).

The increased cell death in miR-126 knockdown embryos was completely suppressed by treatment of miR-126 knockdown embryos with the mTOR activator MHY1485 demonstrating a role for mTOR activity in regulating cell death at the host–mycobacterial interface (Fig 6F).

### miR-126 knockdown increases the migration of macrophages to sites of *M. marinum* infection

As knockdown of miR-126 also increased expression of the neutrophil-related genes, *cxcr4b* and *cxcl12a*, and we had previously found a role for up-regulation of these genes in protection from *M. marinum* infection (Wright et al, 2021), we investigated the effect of miR-126 knockdown on neutrophil motility. Neutrophil responses were first assessed via static imaging of whole-body neutrophil numbers and then by time-lapse imaging of trunk-infected embryos for analysis of neutrophil migration in the transgenic (Tg) *Tg(lyzC:GFP)^{nz117}* neutrophil reporter line. Static imaging revealed no difference in total neutrophil numbers between *M. marinum*–infected control and miR-126 knockdown embryos at either timepoint (Fig 7A). Migration of neutrophils to the site of infection was also not altered between miR-126 knockdown and control infected embryos (Fig 7B). This suggests that despite the increased abundance of transcripts encoding the neutrophil chemotactic genes *cxcr4a/b* and *cxcl12a* in miR-126 knockdown embryos, neutrophil recruitment is unperturbed by miR-126 knockdown in *M. marinum*–infected embryos.

Because of the role of CXCL12 in recruiting CCR2-expressing macrophages and directing their function towards anti-inflammatory activities (Sánchez-Martín et al, 2011; Giri et al, 2020), we next investigated the macrophage response to *M. marinum* infection in miR-126 knockdown embryos. We first estimated total macrophage number in *Tg(mfap4:turquoise)^{xt27}* and *Tg(mfap4:tdtomato)^{xt12}* macrophage reporter lines embryos at baseline and after *M. marinum* infection. Although there was no difference in the total macrophage number between control and miR-126 knockdown embryos in the absence of infection, *M. marinum*–infected miR-126 knockdown embryos had significantly more macrophages than control infected embryos (Fig 7C).

To track the recruitment of macrophages to discrete *M. marinum* lesions, embryos were infected with *M. marinum* via a trunk injection that embeds the bacteria away from the caudal haematopoietic tissue (CHT) (Video 1 and Video 2). Compared with control embryos, miR-126 knockdown embryos had an increased number of macrophages at the site of infection from 5 to 24 hpi (Fig 7D). The increased association of macrophages with *M. marinum* was maintained at 3 dpi in miR-126 knockdown embryos (Fig 7E). There was initially no difference in the number of macrophages present in the CHT at 1 dpi; however, CHT macrophage numbers were

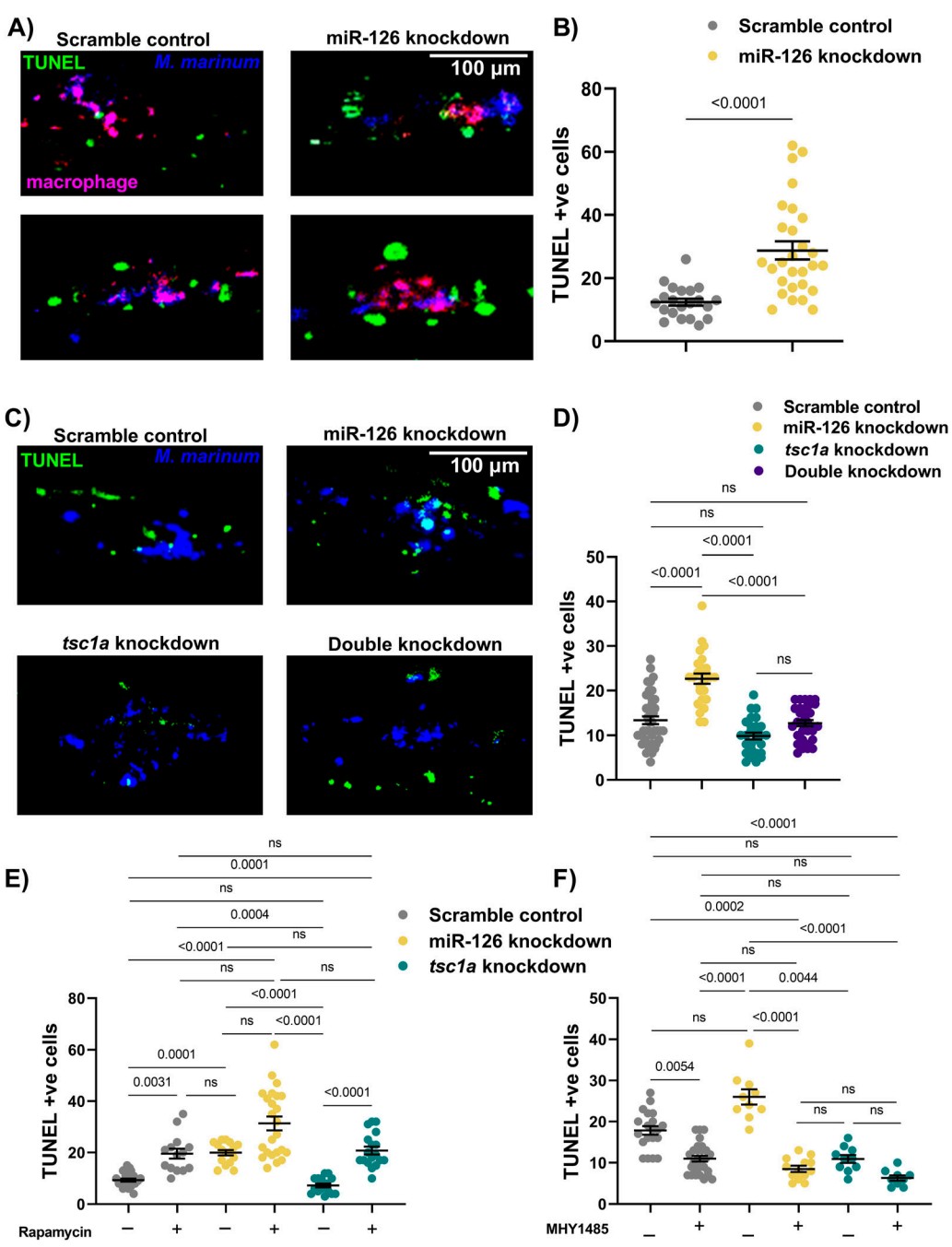

**Figure 6. Decreased mTOR signalling alters cell death dynamics in mycobacterial infection.**
**(A)** Representative images of TUNEL staining in miR-126 knockdown embryos at 3 dpi. TUNEL +ve cells are green, macrophages are magenta, and *M. marinum* is blue. Scale bar represents 100 μm. **(B)** TUNEL +ve cells in *M. marinum*–infected scramble control and miR-126 knockdown embryos were counted at 3 dpi. **(C)** Representative images of TUNEL staining in *tsc1a* and double knockdown embryos at 3 dpi. TUNEL +ve cells are green, and *M. marinum* is blue. Scale bar represents 100 μm. **(D)** TUNEL +ve cells in *M. marinum*–infected *tsc1a* and double knockdown embryos were counted at 3 dpi. **(E)** TUNEL +ve cells in rapamycin-treated, *M. marinum*–infected knockdown embryos were counted at 3 dpi. **(F)** TUNEL +ve cells in MHY1485-treated, *M. marinum*–infected knockdown embryos were counted at 3 dpi. Data information: each data point represents a single measurement, with the mean and SEM shown (n = 20–25 embryos per group). Graphs are representative of two experimental replicates with the exception of rapamycin/MHY1485 experiments, which were performed in a single replicate.

decreased in miR-126 knockdown embryos at 3 dpi, suggesting that reduced miR-126 increases the migration of macrophages but does not influence the production and maturation of immature progenitors (Fig 7F).

To determine if the increased migratory potential of macrophages in miR-126 knockdown embryos was specific to infection, we performed a sterile tail fin wounding assay (Video 3 and Video 4). There was no significant difference in the number of macrophages

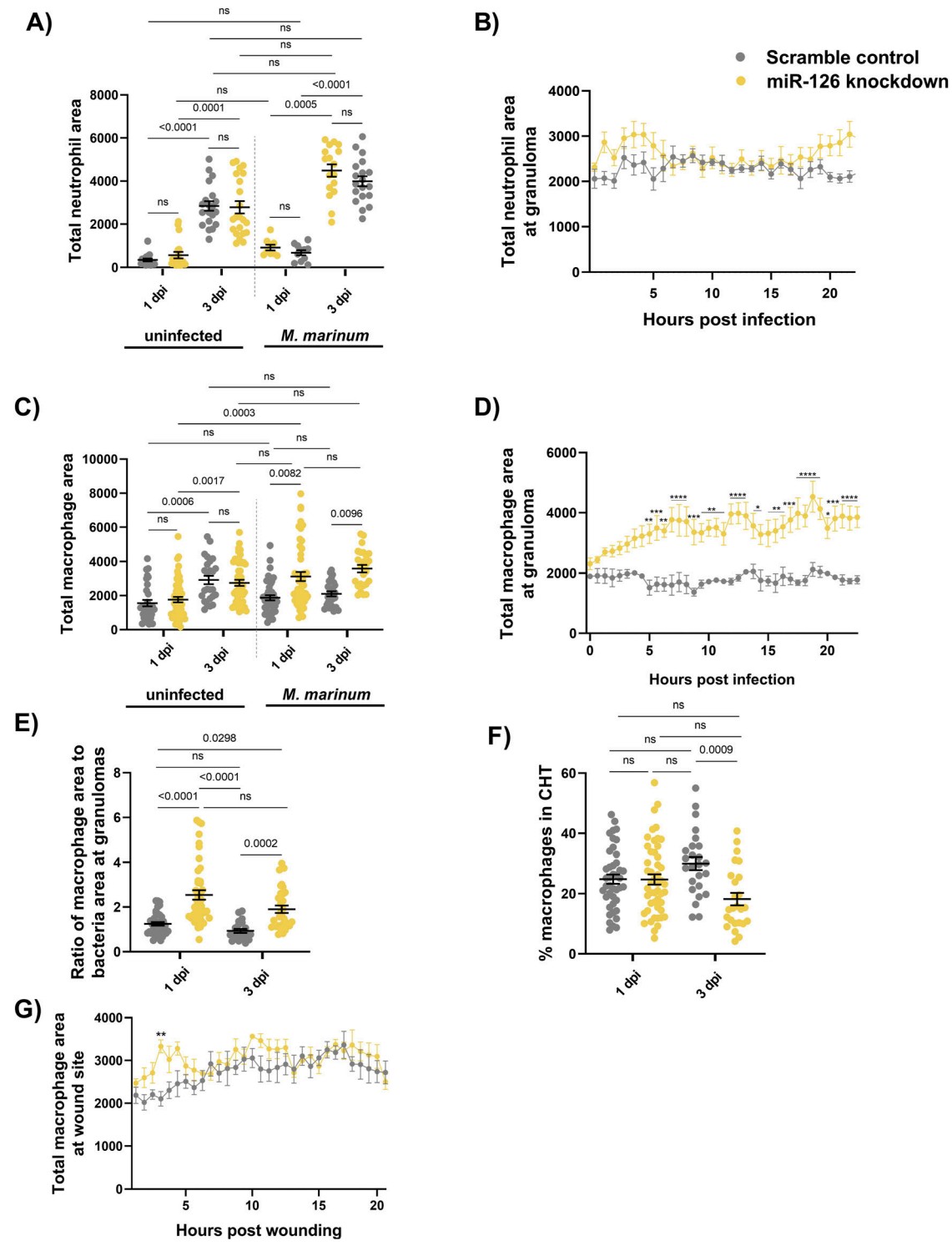

**Figure 7. Mycobacterial infection-induced miR-126 expression alters the host macrophage response.**
**(A)** Measurement of whole-body neutrophil fluorescent area at 1 and 3 dpi in scramble control and miR-126 knockdown uninfected and infected embryos.
**(B)** Measurement of neutrophil levels after trunk infection with *M. marinum* in miR-126 knockdown embryos. **(C)** Measurement of whole-body macrophage fluorescent area at 1 and 3 dpi in uninfected and infected miR-126 knockdown embryos. **(D)** Measurement of macrophage levels after trunk infection with *M. marinum* in miR-126 knockdown embryos. **(E)** Ratio of macrophage fluorescent area per bacterial fluorescent area at granulomas in miR-126 knockdown embryos at 1 and 3 dpi. **(F)** Percent of macrophages residing in the caudal haematopoietic tissue in scramble control and miR-126 knockdown embryos at 1 and 3 dpi. **(G)** Measurement of macrophage recruitment to a tail wound in miR-126 knockdown embryos. Data information: each data point represents a single measurement with the mean and SEM shown. For neutrophil analysis, 10–20 embryos per group were analysed and 15–50 embryos per group for macrophage analysis. For neutrophil time-lapse imaging, each data point

at the wound site between miR-126 knockdown and control embryos, suggesting that the alteration of the macrophage response dynamics is dependent on specific pathogen-derived signals (Fig 7G).

### Increased macrophage recruitment to *M. marinum* infection is independent of Tsc1a/mTOR induction in miR-126 knockdown embryos

We next sought to determine if the Tsc1a/mTOR axis affected macrophage recruitment. After infection with *M. marinum*, *tsc1a* knockdown embryos did not display any difference in total macrophage numbers compared with control embryos at either 1 or 3 dpi (Fig 8A). Double knockdown of miR-126 and *tsc1a* failed to prevent the increased macrophage numbers seen in miR-126 knockdown at 1 dpi, indicating that the altered macrophage response is independent of Tsc1a/mTOR (Fig 8B). To further assess the contribution of mTOR signalling to the observed miR-126–dependant macrophage migration, control and miR-126 knockdown embryos were treated with either MHY1485 or rapamycin and the migratory potential of macrophages measured after trunk infection with *M. marinum* (Fig 8C). As previously observed, miR-126 knockdown embryos exhibited increased macrophage recruitment to sites of infection compared with control embryos. The increased migration of macrophages in miR-126 knockdown embryos was not affected by either activation or inhibition of mTOR by MHY1485 and rapamycin. Likewise, treatment of control embryos with either MHY1485 or rapamycin had no effect on the migration of macrophages to granulomas, suggesting that the increased recruitment of macrophages in miR-126 knockdown embryos is independent of the Tsc1a/mTOR axis.

### Tsc1a depletion increases classical macrophage activation at the site of infection

As a decreased burden was previously observed in *tsc1a* knockdown embryos despite no difference in the macrophage migratory potential and increased inflammatory cytokine transcription by RNAseq, we investigated whether *tsc1a* impacts macrophage activation and phenotype. We next infected *TgBAC(tnfa:gfp)$^{pd1028}$* miR-126 knockdown and *tsc1a* knockdown embryos with *M. marinum* in the trunk region and measured the level of *tnfa* promoter activation around the site of infection (Marjoram et al, 2015). Expression of *tnfa* is a marker of classically activated bactericidal macrophages in zebrafish (Nguyen Chi et al, 2015); we therefore used *tnfa* promoter activation as a surrogate marker of inflammatory, or classically activated, macrophages. At 1 dpi, miR-126 knockdown embryos had significantly reduced activation of the *tnfa* promoter, whereas knockdown of *tsc1a* alone increased *tnfa* promoter activation compared with control and miR-126 knockdown embryos (Fig 8D and E). Double knockdown rescued *tnfa* promoter activation to an

intermediate level, indicating that *tsc1a*-independent mechanisms suppress *tnfa* promoter activation in miR-126 knockdown embryos.

Although the cell death phenotype can be attributed to increased *tsc1a* expression in miR-126 knockdown embryos, and *tsc1a* is partially responsible for the lack of *tnfa* promoter activation, there are clearly *tsc1a*-independent mechanisms responsible for the bulk recruitment of non-*tnfa* promoter active macrophages.

### Knockdown of miR-126 increases expression of *ccl2-ccr2* axis genes associated with permissive macrophages via *cxcl12a* expression

To determine why miR-126 knockdown embryos were unable to contain *M. marinum* in granulomas, despite increased macrophage recruitment, we assessed the activation state of the macrophages. We hypothesised that increased *cxcl12* expression provides more ligand, by heterodimerisation with Ccl2, also known as monocyte chemoattractant protein-1 (Mcp-1), for the recruitment of Ccr2-positive (Ccr2+) permissive or non-bactericidal macrophages to sites of infection (Cambier et al, 2014, 2017).

To investigate this, we first assessed *ccl2* and *ccr2* transcript abundance after infection. Knockdown of miR-126 increased expression of *ccr2* compared with scramble control embryos (Fig 9A). Expression of *ccr2* was also increased when miR-126 knockdown embryos were infected with *M. marinum* compared with infected, scramble control embryos. Although *ccl2* expression was not responsive to either miR-126 knockdown or infection alone, infection of miR-126 knockdown embryos increased *ccl2* expression compared with knockdown alone and *M. marinum*–infected only embryos (Fig 9B). Furthermore, *ccl2* but not *ccr2* expression was observed to be dependent on *cxcl12a* expression, with double knockdown of miR-126 and *cxcl12a* rescuing *ccl2* expression (Fig S3A and B).

Our data thus far indicate that miR-126 knockdown increases the recruitment of macrophages to *M. marinum* infection, and these macrophages are unable to clear bacteria, resulting in macrophage cell death.

### Recruitment of non-*tnfa* expressing macrophages to *M. marinum* infection is dependent on Cxcl12a/Ccr2 signalling in miR-126 knockdown zebrafish embryos

We hypothesised that the increase in expression of Cxcl12/Ccl2/Ccr2 signalling axis components in miR-126 knockdown embryos is responsible for the recruitment of macrophages to infection. To confirm the active role of the Cxcl12/Ccl2/Ccr2 signalling axis as a downstream effector mechanism of miR-126 expression and its involvement in the increased migration of macrophages to sites of mycobacterial infection, the *cxcl12a* and *ccr2* genes were targeted for knockdown using CRISPR-Cas9. Consistent with the work of Cambier et al, we did not observe any effect of *cxcl12a* or *ccr2* knockdown compared with control embryos across the three

---

represents the mean of six foci of infection from six separate embryos, and the graph is representative of two experimental replicates. For macrophage time-lapse imaging, each data point represents the mean of three foci of infection from three separate embryos, and the graph is representative of two experimental replicates. *$P < 0.05$, **$P < 0.01$, ***$P < 0.001$, ****$P < 0.0001$.

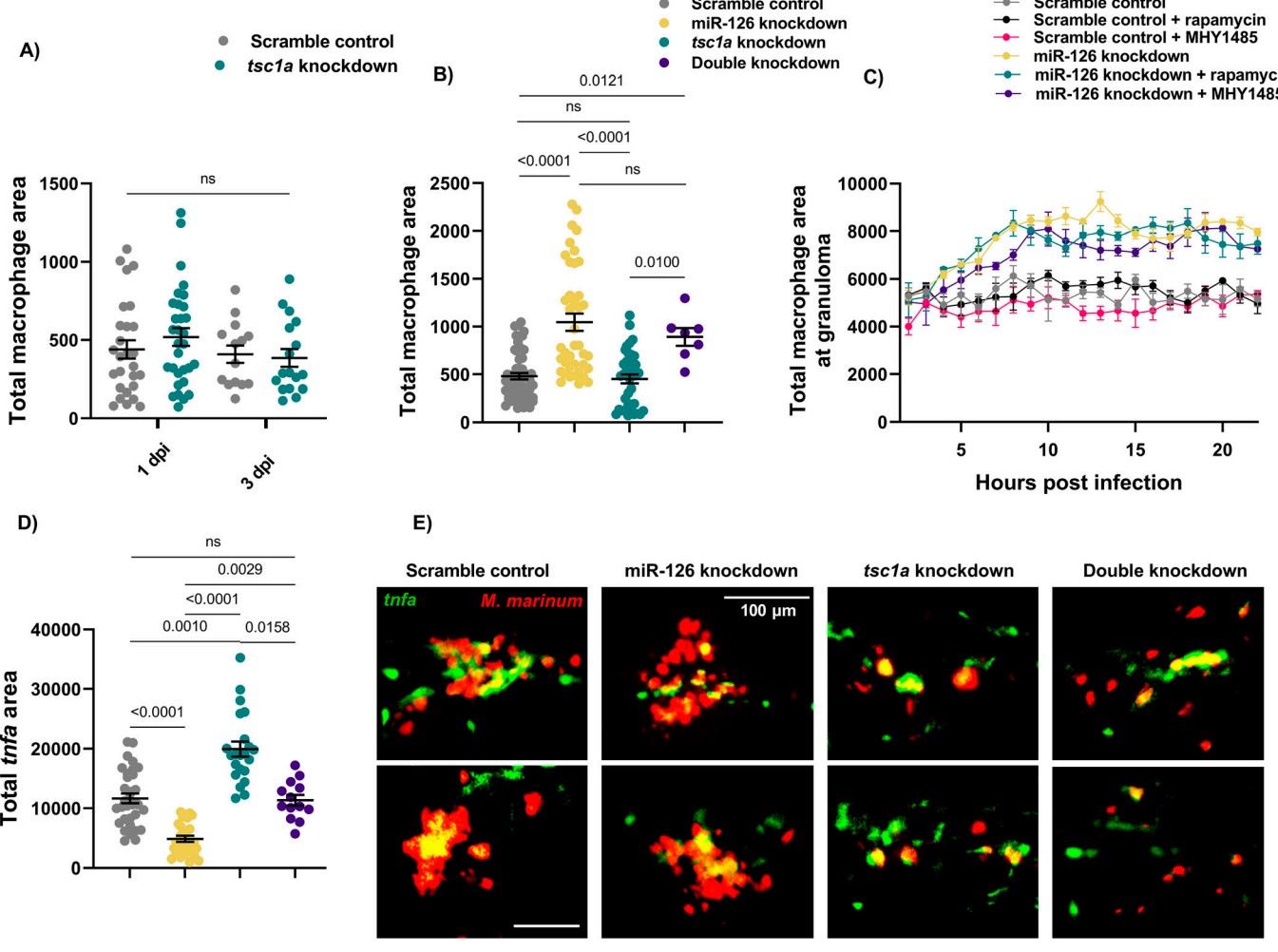

**Figure 8. miR-126–dependent macrophage responses to infection are not controlled by the Tsc1a/mTOR signalling axis.**
**(A)** Measurement of whole-body macrophage fluorescent area at 1 and 3 dpi in *M. marinum*–infected scramble control and *tsc1a* knockdown embryos. **(B)** Measurement of whole-body macrophage fluorescent area at 1 dpi in *M. marinum*–infected scramble control and knockdown embryos. **(C)** Measurement of macrophage levels after trunk infection with *M. marinum* in miR-126 knockdown and control embryos and treatment with either MHY1485 or rapamycin. **(D)** Measurement of *tnfa* promoter activation after trunk infection with *M. marinum* in knockdown embryos at 1 dpi infected with *M. marinum* via caudal vein injection and bacterial burden was analysed at 3 dpi. **(E)** Representative images of *tnfa* promoter-driven GFP expression at 1 dpi in knockdown embryos after trunk infection with *M. marinum*. *M. marinum* is red, and *tnfa* is green. Scale bar represents 100 μm. Data information: each data point represents a single measurement, with the mean and SEM shown (n = 9–56) embryos per group. Graphs are representative of two experimental replicates. For macrophage time-lapse imaging, each data point represents the mean of three foci of infection from three separate embryos.

phenotypes of bacterial burden, total macrophage area, or *tnfa* promoter activity at sites of infection (Fig 10) (Cambier et al, 2014, 2017).

Knockdown of *cxcl12a* and double knockdown of both miR-126 and *cxcl12a* significantly reduced bacterial burden compared with miR-126 knockdown alone (Fig 10A). Similarly, knockdown of *ccr2* and double knockdown of both miR-126 and *ccr2* significantly reduced bacterial burden compared with miR-126 knockdown alone (Fig 10B). There was no difference in bacterial burden between double knockdown embryos and *cxcl12a* or *ccr2* knockdown alone embryos, suggesting the Cxcl12/Ccl2/Ccr2 signalling axis is driving the miR-126 knockdown-associated increase in burden.

Knockdown of *cxcl12a* and double knockdown of both miR-126 and *cxcl12a* significantly reduced total macrophage area compared

with miR-126 knockdown alone (Fig 10C). Similarly, knockdown of *ccr2* and double knockdown of both miR-126 and *ccr2* significantly reduced total macrophage area compared with miR-126 knockdown alone (Fig 10D). Again, there was no difference in total macrophage area between double knockdown embryos and *cxcl12a* or *ccr2* knockdown alone embryos, suggesting that the Cxcl12/Ccl2/Ccr2 signalling axis is driving the miR-126 knockdown-associated increase in macrophage numbers.

Knockdown of *cxcl12a* and double knockdown of both miR-126 and *cxcl12a* restored *tnfa* promoter activation in granulomas to the levels in scramble control embryos (Fig 10E). Similarly, knockdown of *ccr2* and double knockdown of both miR-126 and *ccr2* restored *tnfa* promoter activation in granulomas to scramble control embryos (Fig 10F). As with the other phenotypes, there was no

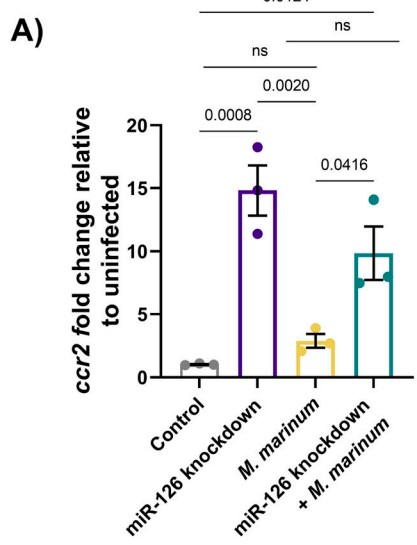

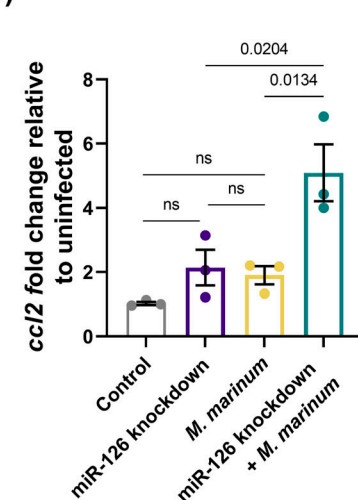

**Figure 9. Mycobacterial infection-induced miR-126 expression increases proinflammatory bactericidal macrophage recruitment.**
**(A, B)** Expression of *ccr2* and *ccl2* was analysed by RT-qPCR at 1 dpi after miR-126 knockdown and infection with *M. marinum*. Expression is relative to uninfected scramble control embryos. Data information: each data point is representative of a single measurement of 10 pooled embryos and two technical replicates, with the mean and SEM shown and comparisons calculated by one-way ANOVA.

difference in *tnfa* promoter activation in granulomas between double knockdown embryos and *cxcl12a* or *ccr2* knockdown alone embryos.

As the increased *ccl2* expression and induction of the Cxcl12a/Ccl2/Ccr2 signalling axis was dependant on *cxcl12a* expression, we examined the migratory potential of macrophages in *cxcl12a* knockdown embryos (Fig 10G). As expected, miR-126 knockdown embryos had more macrophages at the site of infection compared with control embryos, whereas knockdown of *cxcl12a* alone had no effect on macrophage migration. Double knockdown of both miR-126 and *cxcl12a* mitigated the miR-126–dependant increase in macrophage migration, further identifying *cxcl12a* as downstream of miR-126 and as a key regulator of the miR-126/Cxcl12/Ccl2/Ccr2 signalling axis.

Finally, we examined cell death in miR-126 and either *cxcl12a* or *ccr2* knockdown embryos to determine if the lack of *tnfa* expression by recruited macrophages further contributes to the increased cell death observed in miR-126 knockdown embryos (Fig 10H). As previously observed, miR-126 knockdown alone increased cell death compared with control infected embryos. Knockdown of both *cxcl12a* and *ccr2* was protective, decreasing the number of TUNEL-stained cells at 3 dpi. Furthermore, double knockdown of miR-126 and either *cxcl12a* or *ccr2* reduced cell death and from miR-126 knockdown alone, demonstrating that the Cxcl12/Ccl2/Ccr2 signalling axis is driving the miR-126 knockdown-associated increase in ineffective macrophage recruitment to mycobacterial infection.

## Discussion

The ability of mycobacteria to evade host immunity is central to their pathogenicity and ability to establish a chronic infection. Control of miRNA is contested by host and invading mycobacteria as miRNA represents key nodes that are capable of exerting pleiotropic effects on the host immune response. Here, we show

that increased abundance of miR-126 after infection with *M. marinum* is beneficial to the zebrafish host and increases control of the early stages of infection. Our analysis of putative target genes, *tsc1a* and *cxcl12a*, found independent roles of these genes in modulating macrophage activation and death. Our experiments demonstrate that decreasing miR-126 expression early during infection increases the recruitment of a macrophage phenotype which favours infection.

The downstream effects of altered miR-126 expression were mediated in part, by the target gene *tsc1a*. As this gene negatively regulates mTORC1 activity, we were encouraged to find that the miR-126 knockdown phenotype was recapitulated by inhibition of mTOR and restored by co-knockdown of *tsc1a*. However, our data showing mTORC1 activity distal to sites of infection, that inhibition of mTOR further exacerbated bacterial burden and cell death phenotypes in the miR-126 knockdown background, and that the small molecule activation of mTOR only partially rescued the miR-126 knockdown phenotypes suggest that although the miR-126/Tsc1a/mTOR axis is likely, it falls short of fully explaining the effects of miR-126 on the host response to *M. marinum* infection. Although decreased phosphorylation of ribosomal protein S6 was observed in miR-126 knockdown embryos, indicating decreased mTOR activity, further experiments to assess the relative abundance of mTOR targets are required to provide conclusive evidence of a functional miR-126/Tsc1a/mTOR axis.

We observed that inhibition of mTOR with rapamycin is detrimental to the host, enhancing cell death and allowing for uncontrolled growth, whereas small molecule activation of mTOR is beneficial in the zebrafish–*M. marinum* infection model. Previous investigations into the role of mTOR in mycobacterial infection have uncovered a protective role for decreased mTOR signalling through improving mycobacterial killing (Zullo & Lee, 2012; Zullo et al, 2014), so that mTOR inhibitors, such as rapamycin, were considered potential host-directed therapies for the treatment of tuberculosis (Singh & Subbian, 2018). Although our results counter those observed in vitro murine cell culture experiments, they are in

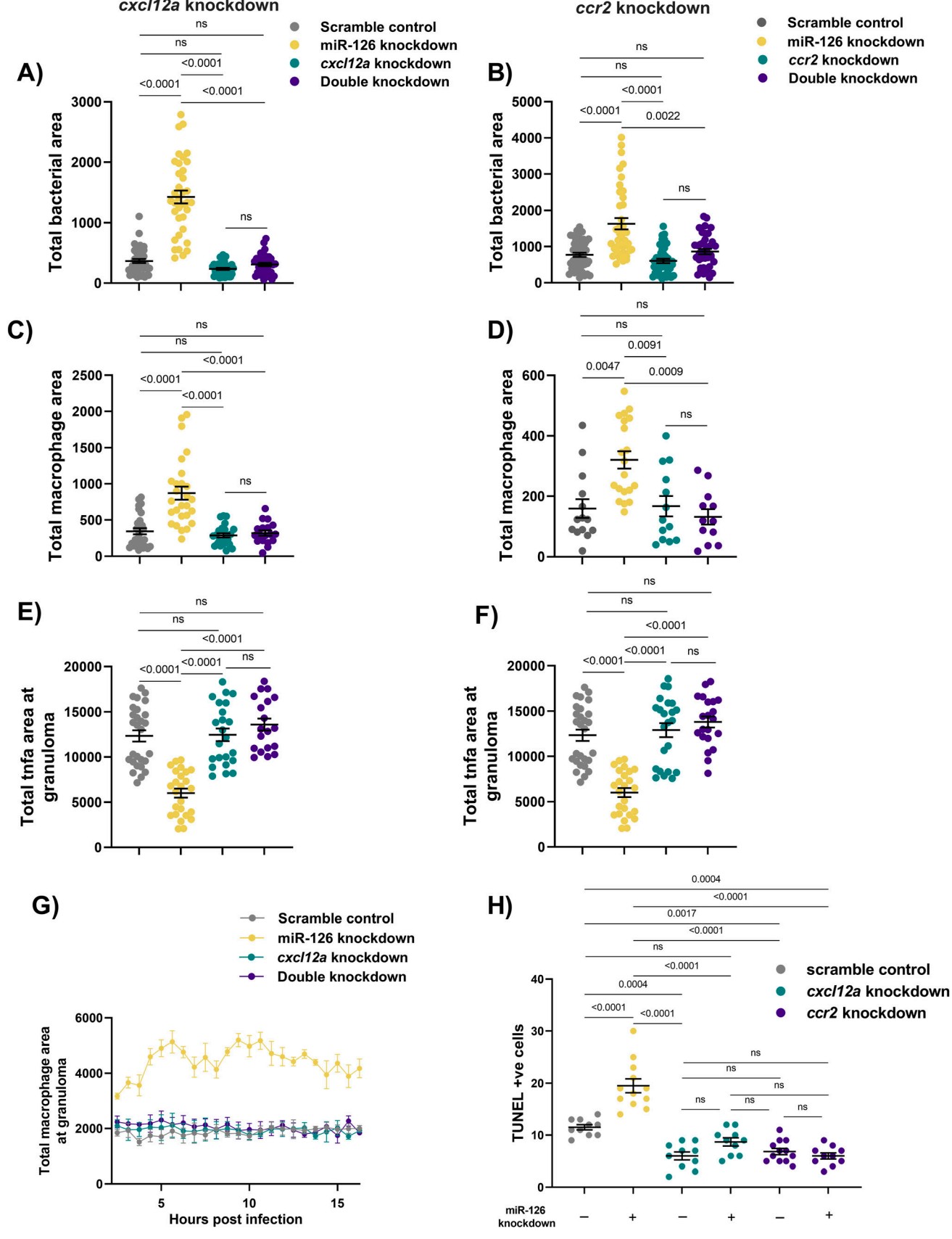

agreement with observations from similar *M. marinum*-zebrafish models, where mTOR deficient zebrafish were more susceptible to mycobacterial infection, displaying severe disease and non-bactericidal macrophages (Pagan et al, 2022). This further reinforces the necessity of in vivo whole organism models of natural mycobacterial infection to capture the complex cellular interactions resulting from alterations of key signalling pathways.

We have previously identified an miRNA-mediated Cxcr4/Cxcl12 signalling axis which increased protective neutrophil responses in mycobacterial infection (Wright et al, 2021). Despite an increase in expression of these same genes after knockdown of miR-126, we found no change in neutrophil recruitment to infection in the current study. This led us to investigate alternative receptors for the Cxcl12 ligand in our miR-126 knockdown embryos that may contribute to the enhanced macrophage influx. Mammalian CXCL12 can form heterodimers with CCL2 to bind to the CCR2 receptor (Giri et al, 2020; Sommer et al, 2021).

Pathogenic mycobacteria use membrane lipids to recruit non-bactericidal Ccr2+ permissive macrophages and ensure bacterial persistence (Antonelli et al, 2010; Cambier et al, 2014, 2017). However, this observation of Ccr2-dependant macrophage recruitment has only been previously observed in zebrafish infected with *M. marinum* in the hindbrain protected by the blood–brain barrier and not in systemic infection despite the up-regulation of *ccl2* after infection of either site (Cambier et al, 2017). In this study, we observed increased bacterial burden associated with the lack of *tnfa:gfp*–positive inflammatory macrophages in the contexts of systemic infection and at localised sites of infection in the trunk musculature only when miR-126 was depleted. The miR-126 knockdown-induced increase in *ccr2* transcription was independent of infection, suggesting that miR-126 is required for the physiological suppression of Ccr2+ macrophage differentiation; thus, the pool of available macrophages in our miR-126 knockdown embryos is potentially skewed away from *tnfa*-expressing macrophages even before infection. We did not examine other markers of macrophage polarisation such as *irg1*, *arg2*, and *nos2a* which would allow further characterisation of the *tnfa:gfp* negative macrophage population in the miR-126 knockdown embryos (Elks et al, 2013; Sanderson et al, 2015; Hammond et al, 2023).

Although we have associated miR-126 with *tsc1* and the *cxcl12a/ccl2/ccr2* axis by combinatorial gene depletion studies, miR-126 has been documented to be involved in additional pathways that may impact the outcome of infection. miR-126 has well characterised roles in the formation of blood vessels and lymphatics (Fish et al, 2008; Zou et al, 2011; Chen et al, 2016; Kontarakis et al, 2018; Qu et al, 2018). Angiogenesis is a major host pathway appropriated by pathogenic mycobacteria to promote pro-angiogenic programmes and increase vascular permeability, enabling bacterial dissemination (Oehlers et al, 2015, 2017; O'Brien J et al, 2018). It is therefore possible that although increased miR-126 is host protective through preventing the recruitment on non-*tnfa* expressing macrophages, pathogenic mycobacteria may co-opt this pathway to increase infection-induced angiogenesis around mature granulomas after the time period studied in our present work. It is evident that miR-126 has numerous target genes involved in a variety of cellular pathways relevant to mycobacterial infection and that its effect on angiogenesis during disease certainly warrants further investigation in a model with established mycobacterial granulomas.

miR-126 has previously been identified as having decreased expression in both plasma from pulmonary tuberculosis patients and PBMCs of tuberculous meningitis patients (Barry et al, 2018; Pan et al, 2019). We have observed that a reduction in miR-126 transcript alters normal macrophage phenotype and function through its involvement in the Cxcl12/Ccl2/Ccr2 axis and coupled with concurrent mTOR inhibition, increases cell death at the site of infection. The lack of bacterial containment and failure to control infection seen in our miR-126 knockdown embryos may provide an insight into the effect of reduced miR-126 transcript abundance in *M. tuberculosis* infection and leads to the question if the reduced miR-126 seen in tuberculosis patients is a host-encoded cause of susceptibility or driven by specific pathogen factors that are not present in our model system.

Importantly, our study falls short of demonstrating direct interactions between miR-126 and the putative target genes specifically by failing to confirm gene expression differences by RNAseq and direct binding of miR-126 to target sequences in the transcripts of putative targets. However, our experimental data points towards strong and consistent functional interactions between miR-126 knockdown and endpoints of macrophage activation/resistance status, so we conclude that we have identified roles for Tsc1a and Cxcl12 in macrophage function as putatively downstream of miR-126 in mycobacterial infection. We demonstrate that *tsc1* and *cxcl12a* expression modulate macrophage function via activation of mTOR signalling and recruitment of *tnfa*-negative macrophages, respectively. Through using a zebrafish–*M. marinum* model, we were able to identify complex multicellular interactions from converging biological pathways that alter the course of mycobacterial infection. These responses appear to be conserved across mycobacterial infection in vertebrate hosts and provide further insight into the intricate regulation of immunity by miRNA.

# Materials and Methods

### Zebrafish husbandry

Adult zebrafish were housed at the Centenary Institute or A*STAR IMCB Aquarium, and breeding was approved by Sydney Local Health

---

**Figure 10. Infection-induced miR-126 regulates Cxcl12/Ccl2/Ccr2 signalling to restrict macrophage recruitment to sites of infection.**
**(A, B)** Bacterial burden measured at 1 dpi in miR-126 and *cxcl12a* (A) or *ccr2* (B) knockdown embryos infected with *M. marinum*. **(C, D)** Whole-body macrophage levels measured at 1 dpi in miR-126 and *cxcl12a* (C) or *ccr2* (D) knockdown embryos infected with *M. marinum*. **(E, F)** *tnfa* fluorescent area at sites of infection measured at 1 dpi in miR-126 and *cxcl12a* (E) and *ccr2* (F) knockdown embryos infected with *M. marinum*. **(G)** Measurement of macrophage levels after trunk infection with *M. marinum* in miR-126 knockdown and *cxcl12a* embryos. **(H)** TUNEL +ve cells counted at 3 dpi in miR-126 and *cxcl12a* or *ccr2* knockdown embryos infected with *M. marinum*. Data information: each data point represents a single measurement, with the mean and SEM shown (n = 10–45) embryos per group. Bacterial burden, macrophage analysis, and granuloma *tnfa* graphs are representative of two experimental replicates, whereas TUNEL staining is presented as a single replicate. For macrophage time-lapse imaging, each data point represents the mean of three foci of infection from three separate embryos.

District (AWC Approval 17-036 and 22-014 or A*STAR IACUC protocol 211667). Embryos were obtained by natural spawning and were raised in E3 media and maintained at 28–32°C.

## Zebrafish lines

Zebrafish were of AB strain. Transgenic lines used were *Tg(lyzC:GFP)*[nz117] and *Tg(lyzC:DsRed2)*[nz50] for neutrophil imaging experiments (Hall et al, 2007), *Tg(mfap4:turquoise)*[xt27] and *Tg(mfap4:tdtomato)*[xt12] for macrophage imaging experiments (Walton et al, 2015), and *TgBAC(tnfa:gfp)*[pd1028] for *tnfa* promotor activation experiments (Marjoram et al, 2015).

## Embryo microinjection with antagomiR

Embryos were obtained by natural spawning and were injected with either miR-126 antagomiR (-GCAUUAUUACUCACGGUACGA-) or a scramble control (-CAGUACUUUUGUGUAGUACAA-) (GenePharma) at 200 pg/embryo at the single-cell stage and maintained at 32°C.

## miRNA target prediction

Prediction of target mRNA was performed using TargetScan. dre-miR-126a-3p was entered into TargetScanFish 6.2 (http://www.targetscan.org/fish_62/), hsa-miR-126-3p entered into TargetScan 7.2 (http://www.targetscan.org/vert_72/), and mmu-miR-126a-3p entered into TargetScanMouse 7.2 (http://www.targetscan.org/mmu_72/). Experimentally validated targets were compiled using miRTarBase (release 8.0) (http://mirtarbase.cuhk.edu.cn). Genes that appeared in multiple software predictions and were experimentally validated as target of miR-126 in either human, mouse, or zebrafish studies were chosen to explore as potential downstream targets of miR-126. Prediction of binding of miR-126 and *tsc1a* in zebrafish was performed using RNAhybrid (https://bibiserv.cebitec.uni-bielefeld.de/rnahybrid) (Rehmsmeier et al, 2004).

## RNA sequencing

Embryos were obtained by natural spawning and were injected with either a miR-126 antagomiR, *tsc1a* CRISPR mixture, or a corresponding control. At 2 dpf, miR-126 knockdown and *tsc1a* knockdown and respective controls were infected with ~200 CFU *M. marinum* via the caudal vein. At 1 dpi (3 dpf), groups of 20 embryos were harvested in TRIzol and RNA extracted as described below. RNA was analysed on a bioanalyzer using the RNA Nano assay (Agilent) to ensure a RIN value ≥ 9 and non-degraded sample RNA. Library preparation and sequencing were performed by Novogene.

mRNA library preparation was performed using the TruSeq RNA Library Prep Kit v2 according to the manufacturer's instruction (Illumina). Sequencing of the mRNA libraries was performed using the NovaSeq platform (Illumina) at Novogene.

The quality of raw sequencing reads was checked using FASTQC (v 0.11.8) (Andrews, 2010). Low-quality reads and adapter sequences were removed to obtain clean reads using TRIMMOMATIC (v 0.39) (Bolger et al, 2014). Clean reads were then mapped to Ensembl zebrafish genome (Danio_rerio.GRCz11.108) using STAR aligner (v 2.7.8a) (Dobin et al, 2013). Samtools (v 1.6) were used to index the aligned bam files (Li et al, 2009). The aligned bam files were quantified with featureCounts (v 2.0.2) to generate read counts of each gene for all samples (Liao et al, 2014). Differential expression analysis was performed using the DESeq2 (v 1.40.2) (Love et al, 2014). Differentially expressed genes were determined using the threshold of an adjusted $P < 0.05$ and $|\log_2 FoldChange| ≥ 1.5$. Gene ontology (GO) enrichment analysis was performed for the significantly increased or decreased genes to identify significantly associated GO terms (an adjusted $P < 0.05$) using clusterProfiler (v 4.8.3) (Yu et al, 2012). Gene Set Enrichment Analysis was performed to identify the significantly enriched gene sets after the tsc1a knockdown (Subramanian et al, 2005). These gene sets were downloaded from the Molecular Signatures Database (MSigDB v7.2; https://www.gsea-msigdb.org/gsea/msigdb/index.jsp) (Liberzon et al, 2011). ES represents enrichment score; NES is the normalised ES value after correction.

## *M. marinum* culture

*M. marinum* was cultured as previously described (Matty et al, 2016). Briefly, *M. marinum* M strain expressing Wasabi, Katushka, or tdTomato fluorescent protein was grown at 28°C in 7H9 supplemented with OADC and 50 µg/ml hygromycin to an OD600 of 007E0.6 before being washed and sheared by aspiration through a 32 G needle into single-cell preparations. These were then aliquoted and frozen in 7H9 at –80°C until needed. The concentration of bacteria was quantified from thawed aliquots by CFU recovery onto 7H10 supplemented with OADC and 50 µg/ml hygromycin and grown at 28°C.

## UPEC culture

Uropathogenic *Escherichia coli* (UPEC) carrying the mCherry PGI6 plasmid (Duggin et al, 2015; Iosifidis & Duggin, 2020) was cultured as previously described (Wright et al, 2021; Tran et al, 2022). Briefly, bacteria were cultured in LB supplemented with 50 µg/ml of spectinomycin overnight at 37°C with shaking at 200 rpm with ~5 cm radius generating ~2 g. Bacteria were then further diluted 1:10 with LB + spectinomycin (50 µg/ml) and incubated for 3 h at 37°C with 200 RPM shaking. Culture medium/broth (1 ml) was centrifuged (16,000*g* for 1 min), and the pellet washed in PBS. The bacterial pellet was resuspended in 300 µl of PBS + 10% glycerol and aliquoted for storage. Enumeration of bacteria was performed by serial dilution on LB + spectinomycin agar plates and culturing at 37°C overnight. Bacterial concentration was determined by CFU counts.

## Bacterial infections

Embryos were dechorionated and anesthetised in tricaine (160 µg/ml) and staged at ~1.5 dpf. Working solutions of *M. marinum* or UPEC (diluted with 0.5% wt/vol phenol red dye) were injected into either the caudal vein or trunk to deliver ~200 CFU *M. marinum* or 250 CFU UPEC. Embryos were recovered in E3 media + PTU (0.036*g*/liter) and maintained at 28°C.

**Table 1. Guide RNA sequences used for CRISPR-Cas9 mediated knockdown experiments.**

| | Primer |
|---|---|
| miR-126 target 1 | TAATACGACTCACTATAGGGCCACAACTGTGTCAGTGTTTTAGAGCTAGAAATAGC |
| miR-126 target 2 | TAATACGACTCACTATAGGCAGTGGGTTTATCTGCGGTTTTAGAGCTAGAAATAGC |
| miR-126 target 3 | TAATACGACTCACTATAGCGTGGACAAACTCAAGACGTTTTAGAGCTAGAAATAGC |
| tsc1a target 1 | TAATACGACTCACTATAGGGTCCTGTAGGCCATGCTCGTTTTAGAGCTAGAAATAGC |
| tsc1a target 2 | TAATACGACTCACTATAGGTCAAATCCTCAGGGATGAGTTTTAGAGCTAGAAATAGC |
| tsc1a target 3 | TAATACGACTCACTATAGGCTGCTGAGGCCTTTCGGAGTTTTAGAGCTAGAAATAGC |
| tsc1a target 4 | TAATACGACTCACTATAGGTGGGCCCTGAGGCGCACAGTTTTAGAGCTAGAAATAGC |
| ccl2 target 1 | TAATACGACTCACTATAGGGAGGACTGTTCCCATCTTGTTTTAGAGCTAGAAATAGC |
| ccl2 target 2 | TAATACGACTCACTATAGGTCAACAAACGCAACCATGGTTTTAGAGCTAGAAATAGC |
| ccl2 target 3 | TAATACGACTCACTATAGGAGCACTTATCGGGACTCTGTTTTAGAGCTAGAAATAGC |
| ccl2 target 4 | TAATACGACTCACTATAGGCAGCACTTATCGGGACTCGTTTTAGAGCTAGAAATAGC |
| ccr2 target 1 | TAATACGACTCACTATAGGGTTGCCATTGTTGCATGGGTTTTAGAGCTAGAAATAGC |
| ccr2 target 2 | TAATACGACTCACTATAGGTTGTCCCATTATTAGCATGTTTTAGAGCTAGAAATAGC |
| ccr2 target 3 | TAATACGACTCACTATAGGAAACGATACTGTACAGGGGTTTTAGAGCTAGAAATAGC |
| ccr2 target 4 | TAATACGACTCACTATAGGTTAGGTAGGGACGCAAACGTTTTAGAGCTAGAAATAGC |
| scramble target 1 | TAATACGACTCACTATAGGCAGGCAAAGAATCCCTGCCGTTTTAGAGCTAGAAATAGC |
| scramble target 2 | TAATACGACTCACTATAGGTACAGTGGACCTCGGTGTCGTTTTAGAGCTAGAAATAGC |
| scramble target 3 | TAATACGACTCACTATAGGCTTCATACAATAGACGATGGTTTTAGAGCTAGAAATAGC |
| scramble target 4 | TAATACGACTCACTATAGGTCGTTTTGCAGTAGGATCGGTTTAGAGCTAGAAATAGC |
| scaffold | AAAAGCACCGACTCGGTGCCACTTTTTCAAGTTGATAACGGACTAGCCTTATTTTAACTTGCTATTTCTAGCTCTAAAAC |

Guide RNA primers used for the generation of template DNA by annealing to the scaffold primer before RNA transcription for injection with Cas9 to generate F0 CRISPR knockdown embryos.

## Bacterial recovery

For CFU recovery, embryos (1.5 dpf) were infected with ~60 CFU fluorescent *M. marinum* into the caudal vein. At 1 and 3 dpi, embryos were first imaged using a fluorescent stereo microscope to measure bacterial burden, followed by immediate collection for CFU recovery. After live imaging of bacterial burden, groups of five embryos were euthanised by a tricaine anaesthetic overdose and homogenised with a 27-gauge needle in 1 ml of PBS. Homogenates were serially diluted in PBS and plated on 7H10 agar quad plates supplemented with OADC and 50 $\mu$g/ml hygromycin. Plates were incubated for 14 d at 28°C, and CFU enumerated using a fluorescent stereo microscope.

## CRISPR-Cas9–mediated knockdown

Embryos were injected at the one- to two-cell stage with 1 nl of CRISPR mixture containing 1 $\mu$g/$\mu$l gRNA, 500 $\mu$g/ml Cas9 (Table 1). For double knockdowns with CRISPR-Cas9 and antagomiR, mixtures contained 1 $\mu$g/$\mu$l gRNA, 100 pg/nl antagomiR, 500 $\mu$g/ml Cas9. gRNA was synthesised as previously described (Wu et al, 2018). Embryos were transferred to E3 containing methylene blue and maintained at 32°C.

## Gene expression analysis

Groups of 10 embryos were lysed and homogenised using a 27-gauge needle in 500 $\mu$l TRIzol (Invitrogen) for RNA extraction. cDNA was synthesised from 500 ng RNA using the miScript II RT Kit with HiFlex buffer. qPCR was carried out on an Mx3000p real-time PCR system using QuantiTect SYBR Green PCR Mastermix (QIAGEN) and primer concentration of 300 nM (Table 2). For miRNA qPCRs, the miScript Universal Primer was used alongside miR-specific miScript primer assays (miR-126 GeneGlobe ID MS00005999 and U6 cat. no. MS00033740 QIAGEN).

Cycling conditions for miRNA were 95°C for 15 min; 40 cycles of 95°C for 20 s, 56°C for 30 s, 72°C for 30 s with fluorescence data acquisition occurring at the end of each cycle, followed by 1 cycle of 95°C for 1 min, 65°C for 30 s, and 97°C for 30 s. For mRNA, conditions were 95°C for 15 min; 40 cycles of 94°C for 15 s, 55°C for 30 s, 70°C for 30 s with fluorescence data acquisition occurring at the end of each cycle, followed by 1 cycle of 95°C for 1 min, 65°C for 30 s, and 97°C for 30 s.

U6 or $\beta$-actin was used as an endogenous control for normalisation, and data analysed using the $2^{-\Delta\Delta}$ Ct method with an average of controls used to set baseline, and fold change was calculated as the ratio of the difference divided by the original value.

## Drug treatment

Embryos were randomly assigned to treatment groups and treated with either 50 nM rapamycin (Sigma-Aldrich) or 5 $\mu$M MHY1485 (Sigma-Aldrich) dissolved in DMSO and refreshed daily.

**Table 2. qPCR primer sequences.**

| qPCR primer | Sequence 5′–3′ | Ensembl ID |
|---|---|---|
| ccl2 forward | GTCTGGTGCTCTTCGCTTTC | ENSDARG00000041835 |
| ccl2 reverse | TGCAGAGAAGATGCGTCGTA | |
| ccr2 forward | GCAACAATGGCAACGCAAAG | OTTDARG00000033131 (si:ch211-207g17.2 gene) |
| ccr2 reverse | GTGAGCCCAGAACGGAAGTG | |
| cxcr4a forward | CAGTTTGGACCGGTACCTCG | ENSDARG00000057633 |
| cxcr4a reverse | CCAGGTGACAAACGAGTCCT | |
| cxcr4b forward | TCGCAGACCTCCTGTTTGTC | ENSDARG00000041959 |
| cxcr4b reverse | CCTTCCCGCAAGCAATTTCC | |
| cxcl12a forward | ATTCGCGAGCTCAAGTTCCT | ENSDARG00000037116 |
| cxcl12a reverse | ATATCTGTGACGGTGGGCTG | |
| spred1 forward | TCTCCCTCCACAGGTTTCCT | ENSDARG00000041449 |
| spred1 reverse | CGCACTTCTTACCGTCCAGT | |
| tsc1a forward | CCTTTCGGAGGGTTGAAGGA | ENSDARG00000026048 |
| tsc1a reverse | CTGCAGGCACAAGACCTTTCA | |
| vegfaa forward | TCCCGACAGAGACACGAAAC | ENSDARG00000045971 |
| vegfaa reverse | TTTACAGGTGAGGGGGTCCT | |
| b-actin forward | CCTTCCAGCAGATGTGGATT | ENSDARG00000037870 |
| b-actin reverse | CACCTTCACCGTTCCAGTTT | |

Primer sequences used for RT-qPCR-based gene expression studies.

## Static imaging and burden analyses

Live imaging was performed on anaesthetised embryos on a depression microscope slide. Images were acquired using a Leica M205FA Fluorescent Stereo Microscope equipped with a Leica DFC365FX monochrome digital camera (Leica Microsystems). Images were analysed using ImageJ software to quantify the fluorescent pixel count, defined as fluorescent signal above a consistent set background determined empirically for each experimental dataset (Matty et al, 2016). Data are presented as total fluorescent area (pixels) above the background level.

For static imaging of granuloma associated macrophages, cells within a 200-µm box surrounding bacterial granulomas were measured and classified as "granuloma-associated macrophages." Expression of GFP in the *tnfa:gfp* line was measured within a 500-µm box around infection foci.

## Macrophage and neutrophil tracking analyses

Time-lapse imaging was performed on a DeltaVision Elite at 28°C (GE). After infection with *M. marinum* into the trunk, embryos were mounted in a 96-well black-walled microplate in 1% low-melting point agarose topped up with E3. Images were captured every 60–240 s for 16–24 h. Analysis was performed using ImageJ software. Briefly, every 10–30 images were analysed for the quantity of neutrophils or macrophages in a 1,000 × 500-µm box around infection foci by quantifying the fluorescent pixel count (total area) at each time point.

## TUNEL cell death staining

Embryos were infected with *M. marinum* via caudal vein injection and analysed for apoptotic cells at 1 and 3 dpi using the Click-iT Plus TUNEL Assay (Thermo Fisher Scientific) according to the manufacturer's protocol. Briefly, embryos were fixed in 10% neutral buffered formalin overnight at 4°C. Embryos were permeabilised with proteinase K (10 µg/ml) for 30 min at room temperature and re-fixed in 10% neutral buffered formalin. edUTP and Alexa Fluor 488 incorporation reactions were performed at 37°C protected from light. Embryos were imaged on a DeltaVision Elite (GE), and TUNEL stained cells counted using the Multi-point Tool in ImageJ.

## Embryo whole-mount immunofluorescence

Embryos were infected with *M. marinum* via caudal vein injection and stained for phosphorylated ribosomal protein S6. At 1 dpi, embryos were fixed in 4% PFA overnight at 4°C followed by several washes with PBS + Tween 20 (PBST), and permeabilisation with proteinase K (10 µg/ml) for 15 min was performed. Embryos were then washed with PBST and refixed in 4% PFA for 20 min. Samples were blocked in 5% goat serum before incubation with the primary antibody (P-S6 Ser235/236 1:100, 4856S; Cell Signalling Technologies) overnight at 4°C with gentle rocking. Embryos were then washed with PBST and blocked with 5% goat serum before addition of the secondary antibody (goat anti-rabbit IgG H&L DyLight 650, 1:200, 84546; Invitrogen) and incubated overnight at 4°C with gentle rocking. Embryos were thoroughly washed with PBST and transferred to a 1:1 PBS/glycerol solution before imaging. Embryos were

imaged on a DM600B (Leica Microsystems), and phospho-S6 quantified using the fluorescent pixel count (total area).

## Statistics

Statistical analysis was performed in GraphPad Prism (v.9.0.0) using default parameters. All data were analysed by a Mann–Whitney test or Kruskal–Wallis non-parametric test depending on the experimental design, and comparisons between groups performed using corrected Dunn's test. Any normally distributed data were analysed with one-way ANOVA and Tukey's multiple comparisons test as indicated in figure legends. For time-lapse data, group comparisons were computed using the Sidak test. Outliers were removed before statistical analysis using ROUT, with Q = 1%.

No sample size estimate or blinding was carried out for this study.

## Data Availability

The RNA sequencing data from this publication have been deposited to the NCBI GEO database under the identifiers GSE253537 (miR-126 KD, Fig S2) and GSE246264 (tsc1a KD, Fig 4).

## Supplementary Information

## Acknowledgements

The authors would like to thank Dr Angela Kurz of the BioImaging Facility and Sydney Cytometry at Centenary Institute for technical assistance with imaging, Drs Pradeep Manuneedhi Cholan and Kaiming Luo for assistance with imaging, and Drs Yilong Lian and Sherry Aw for assistance with additional experiments. We would like to thank Ms Lina Daniel and A/Prof Carl Feng for providing reagents. We would also like to thank Professor Lalita Ramakrishnan and Dr Antonio Pagan for valuable insights. The work was supported by a Meat and Livestock Australia Grant (P.PSH.0813) to K de Silva, KM Plain, and AC Purdie; Higher Degree Research scholarship (P.PSH.0813) to K Wright; Centenary Institute Summer Scholarship to A Shepherd; University of Sydney Fellowship (G197581) and NSW Ministry of Health under the NSW Health Early-Mid Career Fellowships Scheme (H18/31086) to SH Oehlers; and the National Health and Medical Research Council Centre of Research Excellence in Tuberculosis Control (APP1153493) and the University of Sydney DVCR award to WJ Britton.

## Author Contributions

K Wright: conceptualisation, data curation, formal analysis, supervision, funding acquisition, investigation, visualisation, methodology, project administration, and writing—original draft, review, and editing.
DJY Han: data curation, formal analysis, investigation, visualisation, and methodology.
R Song: data curation, software, formal analysis, investigation, visualisation, methodology, and writing—original draft, review, and editing.
K de Silva: conceptualisation, funding acquisition, and writing—review and editing.
KM Plain: conceptualisation, funding acquisition, and writing—review and editing.
AC Purdie: conceptualisation, funding acquisition, and writing—review and editing.
A Shepherd: data curation and investigation.
M Chin: data curation and investigation.
E Hortle: software, formal analysis, supervision, investigation, visualisation, and methodology.
JJ-L Wong: resources, data curation, software, formal analysis, supervision, investigation, visualisation, methodology, and writing—review and editing.
WJ Britton: resources, formal analysis, supervision, funding acquisition, and writing—review and editing.
SH Oehlers: conceptualisation, resources, data curation, formal analysis, supervision, funding acquisition, validation, investigation, visualisation, methodology, project administration, and writing—original draft, review, and editing.

## Conflict of Interest Statement

The authors declare that they have no conflict of interest.

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
