## [Reviewer comments · Life Science Alliance]

Life Science Alliance

Zebrafish *tsc1* and *cxcl12a* increase susceptibility to mycobacterial infection

Kathryn Wright, Darryl Han, Renhua Song, Kumudika de Silva, Karren Plain, Auriol Purdie, Ava Shepherd, Maegan Chin, Elinor Hortle, Justin Wong, Warwick Britton, and Stefan Oehlers

DOI: <https://doi.org/10.26508/lsa.202302523>

Corresponding author(s): *Stefan Oehlers, Agency for Science, Technology and Research*

Review Timeline:

Submission Date:	2023-12-13
Editorial Decision:	2023-12-13
Revision Received:	2024-01-18
Editorial Decision:	2024-01-22
Revision Received:	2024-01-22
Accepted:	2024-01-23

Scientific Editor: *Eric Sawey, PhD*

Transaction Report:

Please note that the manuscript was previously reviewed at another journal and the reports were taken into account in the decision-making process at *Life Science Alliance*.

Referee #1 Review

Report for Author:

The authors have address most, however not all of the concerns, some not even with arguments (which is at least surprising given the long revision time). We appreciate the attempts to improve the paper with RNA sequencing. Due to rather mysterious methodological difficulties, no results could be obtained for sequencing of miR-126 knockdown embryos and the manuscript was reoriented with a focus on the impact on *tsc1* and *cxcl12* instead. Several major issues remain, some of which were already pointed out in the first round of reviews.

1. Although the authors failed to prove the existence of a direct miR-126/*tsc1* axis and realigned the manuscript accordingly, this change in overall direction is not reflected by the current title. This needs to be changes
2. The apparently uninterpretable RNA seq analysis remains a mystery ("very truncated lists of DEGs"). Did the authors fail to extract RNA in sufficient quantity/quality? If so why? What was exactly tried (e.g. scaling up)? Which DEG were identified? This needs to be clarified, i.e. data need to be presented in the supplement.
3. The reviewer asked for the original expression data in uninfected embryos (e.g. relative expression). These data are still missing. Therefore, the data are requested again as supplementary information together with an explanation on how the fold change for uninfected controls was calculated (e.g. in figure 2A-E).
4. The reviewer asked for a further characterization of the macrophages (e.g. by iNOS). Any reply from the authors regarding this aspect is still missing.
5. The quality of the graphical abstract (figure 11) is not suitable for this journal. Please provide a complete, visually appealing graphical abstract.

Additional minor aspects:

Line 45 Please correct: mTOR inhibition. We find

Figure 1 B The legend is missing.

Figure 1 E Please remove statistics where $n = 2$; this is not suitable to perform statistical analysis. Should not Scram inf show around a two-fold increase compared to Scram uninf (compare to 1A)?

Line 373:wq Please correct: Our data thus far indicates

Referee #2 Review

Report for Author:

This manuscript demonstrated that miR-126 upregulation protects the zebrafish embryonic mycobacterium model. Two possible miR-126 targets *Tsc1* and *Cxcl12*, mediate mTOR signaling or macrophage recruitment and activation, respectively. Suppression of either one in miR-126 deficient condition resulted in reduced bacteria burden. Although the importance of mTOR and CCR2 signaling in Mm infection in zebrafish models has been previously reported, the direct connection of these two pathways to miR-126 and its possible target *tsc1* and *cxcl12* is novel. However, the current study's significant weaknesses still need to be addressed.

Major concerns:

1. All experiments should be replicated three times. Power analysis should be used to determine the proper number of samples of zebrafish larvae. Instead of a range of larvae used in each experiment, the exact number should be specified.
2. It would be essential to show the RNA sequencing data and related pathway analysis. Although the selected RNA level did not show any difference in the RNA sequencing experiment, it does not necessarily mean this result is invalid. Understanding the consequence of overall transcript changes upon miR-126 depletion is essential.
3. The manuscript lacks a description or discussion of the expression pattern for miR-126 upon Mm infection. This information is essential to understand which cells, *Tsc1* or *cxcl12*, exert their biological function. An alternative way is to overexpress miR-126 in specific tissue and determine its impact on the infection outcome.
4. Figure 2 is problematic. *tsc1a* is the only one that showed significant upregulation upon miR-126 suppression. The remaining genes did not show significant changes upon Mm infection and miR-126 deletion. I suggest removing the non-significant data.

Minor concerns:

1. Figure 1E. Why would miR-126 sgRNA injection reduce its mRNA level?
2. Figure 4 should include the transcript names and group them in functional groups in A and C to make the data useful.
3. Data of miR-126 target prediction should be better organized using, for example, a Venn diagram to show the overlap of the predicted and validated targets rather than displaying the raw search results.

Referee #3 Review

Report for Author:

The authors have made substantial improvements to the experiments performed, their presentations and the text surrounding their interpretation.

However, in my opinion, much of the work relies on the system of "knocking down" miR-126 in zebrafish alongside targeted deletion (called knockdown) of the putative target Tsc1a. The current interpretation to much of the work based on these experiments is the existence of 2 pathways leading to miR-126's host defence functions during *M. marinum* infection. 1) TSC1 dependent effects on mTOR & 2) TSC1a independent effects on inflammatory macrophage trafficking. My understanding is that miR-126 knockdown animals - treated with an antagomiR - will have higher levels of TSC1a mRNA and protein. And as such, any TSC1a dependent effects of miR-126 inhibition will be absent in animals in which TSC1a gene has been targeted. And for much of the work around mTOR this tracks. And for other effects on macrophage migration, the miR-126 inhibitory effects persist despite TSC1a targeting.

However, the authors have not convinced me that TSC1a levels differ in miR-126 knockdown. The data in current Fig 2 is not the strongest and further RNA-seq did not support it. The authors need to first demonstrate the validity of this system, before making so many substantial claims based upon it. They also need to better explain what is happening in each system - as they currently refer it as knockdown, even though antagomiRs and CRISPR represent different systems.

Secondly, the claim that zebrafish TSC1a would likely be a miR-126 target because of homology and sequence analysis is not a solid foundation for a manuscript based on this. More detailed experiments need to be presented to confirm this.

If this cannot be provided, the data on TSC1 and miR-126 distinct roles in *M. marinum* host defence need to be discussed separately - possibly in 2 different papers

Referee #1 Review

Report for Author:

Wright et al. show an interesting mechanistic link between miRNA-126 signalling and downstream targets in a *M. marinum* zebrafish model. The basic findings are that mycobacteria-induced miR-126 suppresses *tsc1* and *cxcl12a* thereby improving macrophage function via mTOR signalling and the recruitment of TNF-expressing macrophages. The topic is interesting and of general relevance, and the potential of the study is very good. Yet, the work in its current form is incomplete. Thus, a thorough revision seems necessary to support the model as suggested.

Major points:

- Statistics: Data appear by in large not to be normally distributed, i.e. non-parametric tests and parameters are required. The statistical comparisons between conditions are incomplete and not properly depicted. All conditions of each experiment need to be compared and depicted similarly between panels, e.g. by tables below the figures, or by outlining in the legend, which comparisons reveals
- It seems essential to see the original expression data in uninfected embryos, i.e. relative expression in uninfected controls related to house-keeping genes as data (e.g. Fig.1A + B).
- Fig.1: It is stated "For qPCR analysis, each data point represents 10 embryos, and contains 2 biological replicates." The figure shows 4 data points. Does this mean that 2 of these data points are technical replicates and not biological replicates? If this is the case it is not possible to perform statistics. This aspect should be considered throughout the whole manuscript. Please clarify. Any technical replicates should be removed from the statistical analysis and at least one more biological replicate should be added.

- Figure 1d/2c/d etc.: How many embryos were analysed? Was there a power analysis? Why are numbers so variable (Fig. 2c)? In all legends: The exact number of analysed embryos has to be outlined for each condition (e.g. figure 7).
- Fig 2A: The bacterial load (cfu) for dESX as compared to wt *M.marinum* needs to be shown, the mutant is expected to be attenuated.
- Fig. 3 and following: It is not sufficient (and up to date) to base transcriptional changes only on individual genes. RNAseq needs to be performed in embryos after miR-126 treatment (with and without infection) as well as *Tsc1a* ko embryos, and the alterations need to be depicted avoiding unnecessary bias. Thereby downstream genetic targets can be identified to allow for more stringency in interpreting in particular the "double knock out" conditions.
- While it is interesting to use rather robust markers for gene function like "bacterial area" and macrophages the precise intervention by CrispR gene ko and the read-out do not necessarily match in precision. In particular, the parameter "total bacterial area" needs to be backed up by conventional cfu counts.
- Fig.7 B: I am not convinced that "total neutrophil area at granuloma" is not different between scramble control and miR-126 ko late in infection, but total macrophage area at wound site is. This is complicated statistics.
- Fig. 9 C. Since miR-126 in combination with *tsc1a* ko results in a mixed TNF response, the signalling axis is obviously more complicated than proposed. Again, this strongly argues for RNA sequencing analysis after miR-126 treatment and in *tsc1a* ko
- Please provide a cartoon depicting your model of how miR-126, *tsc1a* interact in mycobacteria infection.

Additional points:

- Protective macrophages: This term should be avoided, if protection cannot be demonstrated by these macrophages specifically. It is unclear whether the macrophage area results from the bacterial count or vice versa.
- Classically activated macrophages may be a better term. Moreover, the cells need to be characterized more precisely, at least by staining for iNOS.
- Similarly: Permissive macrophages is a term that is not justified by the data. The wording on this matter should be much more cautious.
- mTOR signalling should be analysed with respect to the influence of rapamycin/MHY185 on macrophage recruitment.

Minor aspects

- Introduction (Lines 72 -78/Lines 89-90): In Lines 89/90 authors claim to analyse the mycobacterium-driven induction of miR-126. In Lines 72-78 they report examples of a reduction of miR-126 in different mycobacterial infections. For the reader this does not become clear at this point. Is miRNA-126 reduction in the examples described then rather a sign of uncontrolled or severe/disseminated forms of infection? It should be specified (for example comment if the infection dose of 200 CFU *M. marinum* in zebrafish is normally cleared or not, and therefore representative of a controlled or disseminated infection) or this aspect should be dealt with in the discussion alone (Lines 443 ff). In general, a comment on the course of *M. marinum* infection in zebrafish would be helpful for the reader to put the 1 day- and 3 day-infection in the right context.
- Regarding the paper Swaim et al (doi: 10.1128/IAI.00887-06) 200 CFU seem to be an intermediate infection dose. Please explain or discuss: Would a high-dose infection of zebrafish lead to a reduction of miRNA-126? Authors claim an important role of miRNA-126 in early time points of infection. Were later time points analyzed? Why were the timepoints and dosages chosen?
- Method section: How did the table in Supplementary Figure 1 led to the selection of the finally analysed downstream targets?
- Table headings are incomplete.
- Line 151 miR-126 instead of miR-12. Please correct.
- Lines 153-154: Figure 3b does not show an increase of *Cxcr4b* in the knockdown embryo, there is a non-significant reduction. This needs to be corrected or explained. Or is Figure 3g referred to in this sentence?
- Line 165/166: Is 3 dpi referred to (3J) or is the infected miR-126 knockdown meant here? In Figure 3e, there is no significant difference between miR-126 knockdown alone and infection alone.
- Figure 4c: The statistical analysis in this figure needs to be clarified (formatting problem).

- Line 216: Authors claim before that *tsc1a* is a negative regulator of mTOR, so knockdown of miR126 would decrease mTOR. Why is there now the opposite described in line 216?
- Line 234: As noticed before the increase in *Cxcr4b* is only visible at day 3 pi (figure 3). This should be elaborated on.
- Figure 7A and 7C: Apparently, in 7A only the situation in infection is analysed. Yet, according to the figure legend, in 7C uninfected animals were also analysed. Where are those depicted? As authors labelled all conditions with dpi (days post infection) in 7C, the labelling should be controlled. Alternatively, the abbreviation dpi could be used as 'days post injection' (line 98) instead. Why are uninfected controls not necessary in 7A but in 7C?
- Line 354: Please correct: Knockdown of down both *cxcl12a* and *ccr2* was protective, decreasing the number of TUNEL-stained cells at 3 dpi.
- Figure legends
 - o Figure 1: Please correct the whole legend, miR-206?
 - o Please clarify: "Bacterial burden analysis data points represent individual embryos (n=40-50 embryos per group) and are representative of 2 biological replicates". If each data point represents an individual embryo, then 40-50 biological replicates are depicted. Is this a representative plot for 2 experiments? In general, all of the data and not representative plots should be depicted (except for imaging of course). Also, for example in Figure 8 the whole data set needs to be depicted.

Referee #2 Review

Report for Author:

This is an exciting story characterizing the biological function of miR-126 in the zebrafish Mm model. The authors showed convincing evidence that miR-126 is protective, specifically in virulent Mm infection. Suppressing miR-126 at the whole organism level resulted in increased cell death, increased macrophage recruitment to the bacteria, reduced macrophage activation, and increased bacterial burden. Through miR-126 target discovery, they found *tsc-1* and *cxcl12a* as the two functional relevant miR-126 targets. *Tsc-1* and related mTOR pathways regulate cell death and macrophage activation. At the same time, *cxcl12a* promotes *ccl2* expression and activates *ccr2* to regulate macrophage recruitment, activation, and cell death. At the same time, the role of mTOR in the zebrafish Mm infection model is known. The link with miR-126 is novel. Regarding *ccr2*, it is not playing a significant role in this model unless miR-126 is depleted. This result is interesting, given that human patients have reduced miR-126 serum levels. However, there are several issues related to the inclusion of the controls or the interpretation of the data which should be addressed before the manuscript can be accepted for publication.

Major concerns:

1. In several places, such as figure 1, the authors used uninjected control for the miR-126 knockdown. Since the knockdown uses antagomir, which is essentially nucleotide. It is important to include the scramble control to rule out possible activation due to nucleotide injection.
2. Whereas using target prediction software or the database of known miR-126 targets can give some clue, a better approach is to use RNA seq to identify miR-126 regulated gene transcription network under the infection condition, which will allow an unbiased comprehensive understanding of miR-126 regulated pathways.
3. The evidence that *tsc1* and *cxcl12a* are the direct miR-126 targets is missing.
4. In figure 5, although increased mTOR activation is evident, however it is not clear in which cells the mTOR is activated. It would be informative to determine whether the pS6 positive cells are macrophages.
5. Same with Figure 6, are the cells with increased TUNEL staining macrophages?
6. From the data presented in Figure 10, looks like all phenotype resulting from miR-126 knockdown can be rescued with *ccr2* knockdown. Will knocking down *ccr2* and *tsc1* together in this situation have additive effects?
7. line 264, the authors concluded that miR-126 does not influence macrophage production. However, the macrophage number in CHT is reduced, which is quite the opposite of the conclusion.
8. Line 312, as mentioned above, the current manuscript lacks evidence that cell death happens in macrophages.

9. line 417-421. The author implied that miR-126 depletion increase ccr2+ macrophage numbers. However, the data presented in this manuscript investigated the ccr2 levels in the entire organism. Is ccr2 only expressed in macrophages?

10. line 432-442, the authors discussed the possible involvement of spred1 as a miR-126 target. however, the data in Figure 3 D, I did not show a significant regulation of spred1 by miR-126 in infection. Therefore, spred1 is not relevant to miR-126 in the current model and this paragraph should be removed or rewritten.

Minor concerns:

1. It would be interesting to figure out the expression pattern of miR-126 during Mm infection. Is miR-126 expressed in the hematopoietic or non-hematopoietic tissues?

2. I would suggest organizing the paper a bit differently. Figure 9CD should be moved to Figure 8-both looking at tsc1 and macrophages.

Referee #3 Review

Report for Author:

This study by Wright et al describes a protective role for microRNA miR-126 in zebrafish infection with the pathogenic mycobacterium, *M. marinum*, which models human tuberculosis. To explain this observation mechanistically, the authors screen for potential mRNA targets of miR-126 involved in the host response to infection. They uncover some interesting targets including a negative regulator of mTOR, Tsc1. Although some of the bacteria-permissive effects of miR-126 ablation are reverse in Tsc1 "knockdown" animals and ablated by Tsc1 disruption, when examining the role of mTOR in this context, mTOR and Tsc1 independent roles are revealed, linked to miR-126. Notably, these include an effect on chemokine signalling, which triggers the recruitment of mycobacteria-permissive macrophages in the absence of miR-126 and facilitates infection. This data is interesting as it reveals a key host defence mechanism to limit this bacteria-favoring response during infection, which could be relevant to human TB, but because of the compounding effects of miR-126 knockdown alongside target gene ablation, it is still unclear which is the central molecular target for miR-126 mediating its protective effects. Further molecular work defining this target would be required to address this & may simplify the key finding and make this study more relevant to the wider molecular biology audience, as at present its relevance and interest may be quite limited to the host/pathogen field. Some major points are outlined below;

TSCL1 ablation model;

The authors describe the targeting of Tsc1; "we targeted tsc1a for knockdown using CRISPR-Cas9 (Figure 4B). Knockdown of tsc1a significantly reduced transcript abundance". Although some sequence information is provided in Table 1, not much extra information is provided to assess what is occurring in this animal, which is crucial to interpretation of much of the subsequent work and observed phenotypes. Is this a knock-out, ie is the coding sequence for Tsc1 removed? Or is there less RNA made/translated into protein in this model? Judging from data in Fig4C, while Tsc1 RNA is still present, there is less in the "Tsc1 knockdown", which is reduced further with *M.marinum* infection. This reviewer wonders if some of the inconsistencies among the different phenotypes measured (pS6-activity, bacterial burden, cell death, macrophage recruitment, TNF promoter activation, CXCL12 induction etc) is because of a tonic level of Tsc1 still present in the "knock-down" which may regulate some of these phenotypes to the same extent as wild-type, but not all? Crucially, although the authors nicely demonstrate mTOR-independent effects of Tsc1 linked to cell death and bacterial dissemination, TSCL1 does not regulate CCR2+ macrophage recruitment - the cells which allow bacterial replication in this model. Would this phenotype be different in a TSCL1-deficient model? A specific zebrafish could be generated where miR-126 targeting of the Tsc1 3'UTR can directly be targeted - using morpholino target protectors or specific editing of the 3'UTR coding sequence? At a minimum, further information on the levels and activity of TSCL1 in the "knockdown" model should be provided to allow better interpretation of this data. However, a clearer way to ablate/disrupt TSCL1 could allow better clarity among some of the observed effects.

Cellular Heterogeneity;

Although Tsc1 targeting reduces some of the effects of miR-126 inhibition, like bacterial survival and cell death, it does not inhibit macrophage recruitment of CCR2+ cells. Is this cell death observed in the Tsc1 knockdown animal occurring in a different cellular population than the CCR2+ recruited cells, which are recruited in response to the cell death? And is it likely then that miR-126 controls expression of different targets in both macrophage/cellular populations? TSCL1 in the "resident" population and a separate target linked to chemokine signalling in the CCR2+ cells?

Direct miR-126 Targets;

This links to my final major point. The authors use bioinformatic analysis to predict miR-126 targets involved in the host response to mycobacterial infection in the zebrafish model. An initial mRNA screen validates some of these. Tsc1 emerges as a strong candidate and an RNA-Hybrid plot is provided to illustrate the potential interaction. 3'UTR Luciferase Assays should be employed to confirm this, in a reporter cell system. Additionally, it is unclear from the manuscript if CXCL12a or CCL2/CCR2 are thought to be direct targets of miR126 or their expression is regulated by other targets (the Tsc1/mTOR axis?). 3'UTR Luciferase assays would strengthen this and may point to the most relevant targets for miR-126.

Minor points for correction/clarification;

Intro; line 58/59; would it be better to describe microRNAs as "fine-tuners" rather than "master regulators"

Line 72/73; any information on which cells in cattle decreased miR-126 was observed in

Line 75/76; similarly, what cells or tissues in humans?

Results, Fig 7c: in the recruited macrophage reporter mouse, the trace does not appear to show increased recruitment in control (non-knockdown) mice, is this expected?

Remarks To Editors;

This is a well performed and interesting study in a model of infection relevant to human disease, but which can be more easily targeted to reveal novel mechanistic insight. In this case, much has been revealed about miR-126 biology, however, some of this work remains incomplete - particularly the observed Tsc1-independent effects on macrophage recruitment, which seem to be central to the protective effects of miR-126/bacteria-permissive effects of miR-126 inhibition. the distinctions between these 2 pathways need to be more clearly delineated and the following points should be addressed in any potential revision for this journal.

- Clarification on the role of TSCL1 and methods to ablate
- More specific ablation or targeting of the miR-126/TSCL1 interaction would be desirable
- 3'UTR Target Analysis to define direct miR-126 molecular targets
- Is there differences in miR126 targeting in different macrophage/cellular populations?

Referee #1:

Wright et al. show an interesting mechanistic link between miRNA-126 signalling and downstream targets in a *M. marinum* zebrafish model. The basic findings are that mycobacteria-induced miR-126 suppresses *tsc1* and *cxcl12a* thereby improving macrophage function via mTOR signalling and the recruitment of TNF-expressing macrophages. The topic is interesting and of general relevance, and the potential of the study is very good. Yet, the work in its current form is incomplete. Thus, a thorough revision seems necessary to support the model as suggested.

Major points:

- Statistics: Data appear by in large not to be normally distributed, i.e. non-parametric tests and parameters are required. The statistical comparisons between conditions are incomplete and not properly depicted. All conditions of each experiment need to be compared and depicted similarly between panels, e.g. by tables below the figures, or by outlining in the legend, which comparisons reveals

Non-normally disturbed data have been re-analysed using non-parametric tests and all statistical comparisons displayed on each graph. The methods sub-heading "statistics" has been updated to include precise tests used.

- It seems essential to see the original expression data in uninfected embryos, i.e. relative expression in uninfected controls related to house-keeping genes as data (e.g. Fig.1A + B).
- Fig.1: It is stated "For qPCR analysis, each data point represents 10 embryos, and contains 2 biological replicates." The figure shows 4 data points. Does this mean that 2 of these data points are technical replicates and not biological replicates? If this is the case it is not possible to perform statistics. This aspect should be considered throughout the whole manuscript. Please clarify. Any technical replicates should be removed from the statistical analysis and at least one more biological replicate should be added.

Biological replicates are now plotted across all qPCR analyses.

- Figure 1d/2c/d etc.: How many embryos were analysed? Was there a power analysis? Why are numbers so variable (Fig. 2c)? In all legends: The exact number of analysed embryos has to be outlined for each condition (e.g. figure 7).

Number of embryos is now stated in legends. No power analysis was performed.

To be frankly honest if we could make the infections less variable, we would. We assume the zebrafish embryo yields more variable burden data than cell culture and mouse experiments because they are genetically diverse (not inbred) and there is additional variation introduced by the micromanipulation procedure to inject bacteria.

- Fig 2A: The bacterial load (cfu) for dESX as compared to wt *M.marinum* needs to be shown, the mutant is expected to be attenuated.

Yes, the mutant is attenuated. Since we do not have matched infection burdens we have removed Figure 2 entirely.

- Fig. 3 and following: It is not sufficient (and up to date) to base transcriptional changes only on individual genes. RNAseq needs to be performed in embryos after miR-126 treatment (with and without infection) as well as *Tsc1a* ko embryos, and the alterations need to be depicted avoiding unnecessary bias. Thereby downstream genetic targets can be identified to allow for more stringency in interpreting in particular the "double knock out" conditions.

RNAseq analysis of *Tsc1a* KD showed an increase in ribosomal biogenesis consistent with relieving suppression of TOR and an increase in immune gene expression consistent with our phenotype of reduced bacterial burden. This data is presented in Figure 4 and described in the results section around Line 200.

We made two attempts at RNAseq analysis of miR-126 knockdown embryos which generated only very truncated lists of DEGs. Surprisingly despite using RNA samples (generated from two countries) that showed differential expression of *tsc1a*, *cxcr4b*, and *cxcl12a* by our conventional cDNA synthesis/qPCR measurement, these transcripts were all found to be present at equivalent levels by RNAseq measurement. In light of this discrepancy we have rewritten the initial sections to include mention of the uncertainty of a miR-126/*tsc1a* axis and to refocus the writing of the overall manuscript to focus on the downstream *tsc1a* and *cxcl12a* phenotypes. See lines 172 and 500 for specific mention of this failure.

- While it is interesting to use rather robust markers for gene function like "bacterial area" and macrophages the precise intervention by CrispR gene ko and the read-out do not necessarily match in precision. In particular, the parameter "total bacterial area" needs to be backed up by conventional cfu counts.

New data is presented in Supplementary Figure 1 showing a replication of the critical experiment from Figure 1D with measurements by fluorescent pixel count and CFU recovery demonstrating the same pattern of bacterial burden across both measurement techniques. We also direct the reviewer to Walton et al, Cell Host Microbe 2018 (PMID: 30308157) for data showing strong correlation of fluorescent pixel count burden estimation with RT-qPCR measurement of bacterial 16S rRNA as a separate recent validation of fluorescent pixel count compared to a third measurement technique.

- Fig.7 B: I am not convinced that "total neutrophil area at granuloma" is not different between scramble control and miR-126 ko late in infection, but total macrophage area at wound site is. This is complicated statistics.

Although there may in fact be a measurable difference at that particular timepoint, this difference is not statistically significant by the, admittedly simple, Sidak correction ANOVA test that we utilised. We are open to the reviewer's suggestion for a more appropriate test to perform multiple comparisons across treatment groups and along a timeseries.

- Fig. 9 C. Since miR-126 in combination with *tsc1a* ko results in a mixed TNF response, the signalling axis is obviously more complicated than proposed. Again, this strongly argues for RNA sequencing analysis after miR-126 treatment and in *tsc1a* ko

This project has been unfunded since the end of 2021 and additional RNAseq validation of qPCR results is not possible.

- Please provide a cartoon depicting your model of how miR-126, *tsca* interact in mycobacteria infection.

We have added a cartoon as Figure 11.

Additional points:

- Protective macrophages: This term should be avoided, if protection cannot be demonstrated by these macrophages specifically. It is unclear whether the macrophage area results from the bacterial count or vice versa.
- Classically activated macrophages may be a better term. Moreover, the cells need to be characterized more precisely, at least by staining for iNOS.
- Similarly: Permissive macrophages is a term that is not justified by the data. The wording on this matter should be much more cautious.

Wording has been revised to "*tnfa* expressing" to reflect the technical assay used in this study.

- mTOR signalling should be analysed with respect to the influence of rapamycin/MHY185 on macrophage recruitment.

The macrophage migration assay was repeated with rapamycin/MHY1485 treatment demonstrating the increased migration phenotype is mTOR-independent (Figure 8).

Minor aspects:

- Introduction (Lines 72 -78/Lines 89-90): In Lines 89/90 authors claim to analyse the mycobacterium-driven induction of miR-126. In Lines 72-78 they report examples of a reduction of miR-126 in different mycobacterial infections. For the reader this does not become clear at this point. Is miRNA-126 reduction in the examples described then rather a sign of uncontrolled or severe/disseminated forms of infection? It should be specified (for example comment if the infection dose of 200 CFU *M. marinum* in zebrafish is normally cleared or not, and therefore representative of a controlled or disseminated infection) or this aspect should be dealt with in the discussion alone (Lines 443 ff). In general, a comment on the course of *M. marinum* infection in zebrafish would be helpful for the reader to put the 1 day- and 3 day-infection in the right context.

Added a description of our model as “acute disseminated mycobacterial infection” in introduction (line 127) and discrepancy between experimental infection increase in zebrafish miR-126 vs reduced human miR-126 in tuberculosis in discussion (line 623).

- Regarding the paper Swaim et al (doi: 10.1128/IAI.00887-06) 200 CFU seem to be an intermediate infection dose. Please explain or discuss: Would a high-dose infection of zebrafish lead to a reduction of miRNA-126? Authors claim an important role of miRNA-126 in early time points of infection. Were later time points analyzed? Why were the timepoints and dosages chosen?

200 CFU is used as a standard dose for zebrafish embryos (doi: 10.1093/infdis/jiw355 & doi: 10.1038/nature13967) which results in a progressive granulomatous infection. The infection experiments performed by Swaim et. al are performed on adult zebrafish and injected intraperitoneally. Our infection model utilises lower doses due to the size and age of the embryos, as well as the injection of bacteria directly into the circulation or musculature.

Later timepoints could not be analysed for our original experimental set up as the antagomir diluted out after 4-5 days post fertilisation / 3-4 days post infection.

- Method section: How did the table in Supplementary Figure 1 led to the selection of the finally analysed downstream targets?

The methods subsection “miRNA target prediction” has been updated to include selection criteria for downstream targets.

- Table headings are incomplete.

Table headings have been extended.

- Line 151 miR-126 instead of miR-12. Please correct.

Corrected.

- Lines 153-154: Figure 3b does not show an increase of Cxcr4b in the knockdown embryo, there is a non-significant reduction. This needs to be corrected or explained. Or is Figure 3g referred to in this sentence?

Wording has been updated to clarify this... “From the gene expression analysis, *cxcr4b*, *cxcl12a*, and *tsc1a* were regarded as potential target genes of miR-126 due to their increased expression in knockdown and knockdown infected embryos and differential regulation compared to *M. marinum* infected embryos. Despite no increase in expression at 1 dpi of *cxcr4b* in miR-126 knockdown alone, the increased expression in miR-126 knockdown infected embryos made *cxcr4b* a gene of interest alongside *cxcr4a* and *cxcl12a* due to their previously identified role in mycobacterial and zebrafish immunity...”

- Line 165/166: Is 3 dpi referred to (3J) or is the infected miR-126 knockdown meant here? In Figure 3e, there is no significant difference between miR-126 knockdown alone and infection alone.

Text has been corrected and updated to clarify significant differences.

- Figure 4c: The statistical analysis in this figure needs to be clarified (formatting problem).

Formatting of statistical comparisons has been corrected.

- Line 216: Authors claim before that *tsc1a* is a negative regulator of mTOR, so knockdown of miR126 would decrease mTOR. Why is there now the opposite described in line 216?

Incorrect wording (increased mTOR activity) has been corrected to “decreased mTOR activity”.

- Line 234: As noticed before the increase in *Cxcr4b* is only visible at day 3 pi (figure 3). This should be elaborated on.

- Figure 7A and 7C: Apparently, in 7A only the situation in infection is analysed. Yet, according to the figure legend, in 7C uninfected animals were also analysed. Where are those depicted? As authors labelled all conditions with dpi (days post infection) in 7C, the labelling should be controlled. Alternatively, the abbreviation dpi could be used as 'days post injection' (line 98) instead. Why are uninfected controls not necessary in 7A but in 7C?

Labelling has been updated to include designation of uninfected and infected embryos and improve readability of graphs. Uninfected controls have been included for Figure 7A.

- Line 354: Please correct: Knockdown of down both *cxcl12a* and *ccr2* was protective, decreasing the number of TUNEL-stained cells at 3 dpi.

Corrected

- Figure legends
 - o Figure 1: Please correct the whole legend, miR-206?

Legend has been corrected

- o Please clarify: "Bacterial burden analysis data points represent individual embryos (n=40-50 embryos per group) and are representative of 2 biological replicates". If each data point represents an individual embryo, then 40-50 biological replicates are depicted. Is this a representative plot for 2 experiments? In general, all of the data and not representative plots should be depicted (except for imaging of course). Also, for example in Figure 8 the whole data set needs to be depicted.

Wording has been corrected from biological replicates to experimental replicates.

Referee #2:

This is an exciting story characterizing the biological function of miR-126 in the zebrafish Mm model. The authors showed convincing evidence that miR-126 is protective, specifically in virulent Mm infection. Suppressing miR-126 at the whole organism level resulted in increased cell death, increased macrophage recruitment to the bacteria, reduced macrophage activation, and increased bacterial burden. Through miR-126 target discovery, they found *tsc-1* and *cxcl12a* as the two functional relevant miR-126 targets. Tsc-1 and related mTOR pathways regulate cell death and macrophage activation. At the same time, *cxcl12a* promotes *ccl2* expression and activates *ccr2* to regulate macrophage recruitment, activation, and cell death. At the same time, the role of mTOR in the zebrafish Mm infection model is known. The link with miR-126 is novel. Regarding *ccr2*, it is not playing a significant role in this model unless miR-126 is depleted. This result is interesting, given that human patients have reduced miR-126 serum levels. However, there are several issues related to the inclusion of the controls or the interpretation of the data which should be addressed before the manuscript can be accepted for publication.

Major concerns:

1. In several places, such as figure 1, the authors used uninjected control for the miR-126 knockdown. Since the knockdown uses antagomir, which is essentially nucleotide, it is important to include the scramble control to rule out possible activation due to nucleotide injection.

All control embryos used for miR-126 antagomiR knockdown and Crispr knockdown experiments were injected with a corresponding scramble control sequence. Figure legends have been updated to clarify this point.

Additional experimental data using a scramble gRNA/Cas9 vs miR-126 gRNA/Cas9 injected F0 crispant system have been added in Figure 1E and F demonstrating a second line of evidence that miR-126 is host-protective against *M. marinum* infection.

2. Whereas using target prediction software or the database of known miR-126 targets can give some clue, a better approach is to use RNA seq to identify miR-126 regulated gene transcription network under the infection condition, which will allow an unbiased comprehensive understanding of miR-126 regulated pathways.

3. The evidence that *tsc1* and *cxcl12a* are the direct miR-126 targets is missing.

Both *Tsc1* and *Cxcl12a* have previously been identified as direct targets of miR-126. A study in zebrafish has shown direct targeting of *Cxcl12a* by miR-126a (doi: 10.1161/ATVBAHA.116.308120) while multiple studies have identified *Tsc1* as a target of miR-126 in mice, humans, and pigs (doi: 10.1038/ni.2767) (doi: 10.1007/s11626-018-0292-0). As the degree of sequence homology and apparent conservation of target genes is high, we believe that *Tsc1* is a direct target of miR-126 in zebrafish. This has been further mentioned in the text.

We made two attempts at RNAseq analysis of miR-126 knockdown embryos which generated only very truncated lists of DEGs. Surprisingly despite using RNA samples (generated from two countries) that showed differential expression of *tsc1a*, *cxcr4b*, and *cxcl12a* by our conventional cDNA synthesis/qPCR measurement, these transcripts were all found to be present at equivalent levels by RNAseq measurement.

Furthermore, we were unable to perform experiments showing direct binding of zebrafish miR-126 to anything in cell culture transfection experiments using the predicted target sequence as bait in dual luciferase assays.

In light of these failures we have rewritten the initial sections to include mention of the uncertainty of a miR-126/*tsc1a* axis and to refocus the writing of the overall manuscript to focus on the downstream *tsc1a* and *cxcl12a* phenotypes.

4. In figure 5, although increased mTOR activation is evident, however it is not clear in which cells the mTOR is activated. It would be informative to determine whether the pS6 positive cells are macrophages.

We have assessed the distribution and localisation of pS6 staining in macrophage reporter lines showing minimal colocalisation or specificity to macrophages.

5. Same with Figure 6, are the cells with increased TUNEL staining macrophages?

We have included images of TUNEL staining in macrophage reporter lines showing colocalisation of TUNEL positive cells with bacteria and macrophages.

6. From the data presented in Figure 10, looks like all phenotype resulting from miR-126 knockdown can be rescued with *ccr2* knockdown. Will knocking down *ccr2* and *tsc1* together in this situation have additive effects?

This is an interesting idea to look at interaction between the pathways. We attempted to perform combinations of triple knockdown of miR-126/*ccr2*/*tsc1a*. Unfortunately, our embryos did not develop normally which we think is due to too much injected material (antagomir and gRNA/Cas9 complexes). Our live imaging data in Figure 8 provides evidence of independence as manipulation of *tsc1*/mTOR did not affect macrophage migration phenotypes in the control and miR-126 knockdown backgrounds.

7. line 264, the authors concluded that miR-126 does not influence macrophage production. However, the macrophage number in CHT is reduced, which is quite the opposite of the conclusion.

Our data shows reduced macrophage numbers in the CHT only in the context of infection where there is increased recruitment from the CHT to the sites of infection. Thus we hypothesise that rather than decreased production of macrophages, and that the production of new cells is not able to “keep up” with the recruitment requirements (Pagan 2015 Cell Host and Microbe PMID: 26159717).

A detailed study of the role of miR-126 in myelopoiesis/monocyte differentiation will be grounds for future study. We have updated the text to clarify that there is in fact a difference in the numbers and explain our theory.

8. Line 312, as mentioned above, the current manuscript lacks evidence that cell death happens in macrophages.

As above, we have included images with macrophages, bacteria, and TUNEL staining.

9. line 417-421. The author implied that miR-126 depletion increase *ccr2*⁺ macrophage numbers. However, the data presented in this manuscript investigated the *ccr2* levels in the entire organism. Is *ccr2* only expressed in macrophages?

According to a publicly available single cell RNAseq dataset of dissected granulomas from Cronan et al Cell 2021, *ccr2* is expressed broadly by macrophages and dendritic cells. Expression was absent in neutrophils, B and T cells, and RBCs.

We have attempted to perform whole mount in situ hybridisation with our existing reagents to confirm these datasets but were unable to observe clean staining with our probes.

10. line 432-442, the authors discussed the possible involvement of *spred1* as a miR-126 target. however, the data in Figure 3 D, I did not show a significant regulation of *spred1* by miR-126 in infection. Therefore, *spred1* is not relevant to miR-126 in the current model and this paragraph should be removed or rewritten.

Discussion relating to *Spred1* has been removed.

Minor concerns:

1. It would be interesting to figure out the expression pattern of miR-126 during Mm infection. Is miR-126 expressed in the hematopoietic or non-hematopoietic tissues?

miR-126 appears to be expressed fairly ubiquitously. For instance we saw mTOR activity effects in stromal cells in addition to the effects on haematopoietic cell phenotypes which we have focused on and the blood vessel phenotypes documented in the literature.

2. I would suggest organizing the paper a bit differently. Figure 9CD should be moved to Figure 8-both looking at *tsc1* and macrophages.

The additional mTOR modulation data in Figure 8 makes the new Figure 8 quite large already.

Referee #3:

This study by Wright et al describes a protective role for microRNA miR-126 in zebrafish infection with the pathogenic mycobacterium, *M. marinum*, which models human tuberculosis. To explain this observation mechanistically, the authors screen for potential mRNA targets of miR-126 involved in the host response to infection. They uncover some interesting targets including a negative regulator of mTOR, *Tsc1*. Although some of the bacteria-permissive effects of miR-126 ablation are reverse in *Tsc1* "knockdown" animals and ablated by *Tsc1* disruption, when examining the role of mTOR in this context, mTOR and *Tsc1* independent roles are revealed, linked to miR-126. Notably, these include an effect on chemokine signalling, which triggers the recruitment of mycobacteria-permissive macrophages in the absence of miR-126 and facilitates infection. This data is interesting as it reveals a key host defence mechanism to limit this bacteria-favoring response during infection, which could be relevant to human TB, but because of the compounding effects of miR-126 knockdown alongside target gene ablation, it is still unclear which is the central molecular target for miR-126 mediating its protective effects. Further molecular work defining this target would be required to address this & may simplify the key finding and make this study more relevant to the wider molecular biology audience, as at present its relevance and interest may be quite limited to the host/pathogen field. Some major points are outlined below;

TSCL1 ablation model:

The authors describe the targeting of *Tsc1*; "we targeted *tsc1a* for knockdown using CRISPR-Cas9 (Figure 4B). Knockdown of *tsc1a* significantly reduced transcript abundance". Although some sequence information is provided in Table 1, not much extra information is provided to assess what is occurring in this animal, which is crucial to interpretation of much of the subsequent work and observed phenotypes. Is this a knock-out, ie is the coding sequence for *Tsc1* removed? Or is there less RNA made/translated into protein in this model? Judging from data in Fig4C, while *Tsc1* RNA is still present, there is less in the "*Tsc1* knockdown", which is reduced further with *M. marinum* infection. This reviewer wonders if some of the inconsistencies among the different phenotypes measured (pS6-activity, bacterial burden, cell death, macrophage recruitment, TNF promoter activation, CXCL12 induction etc) is because of a tonic level of *Tsc1* still present in the "knock-down" which may regulate some of these phenotypes to the same extent as wild-type, but not all? Crucially, although the authors nicely demonstrate mTOR-independent effects of *Tsc1* linked to cell death and bacterial dissemination, TSCL1 does not regulate CCR2+ macrophage recruitment - the cells which allow bacterial replication in this model. Would this phenotype be different in a TSCL1-deficient model? A specific zebrafish could be generated where miR-126 targeting of the *Tsc1* 3'UTR can directly be targeted - using morpholino target protectors or specific editing of the 3'UTR coding sequence? At a minimum, further information on the levels and activity of TSCL1 in the "knockdown" model should be provided to allow better interpretation of this data. However, a clearer way to ablate/disrupt TSCL1 could allow better clarity among some of the observed effects.

The use of a *tsc1a* null mutant would be the ideal response to this concern. Our additional data in Figure 4 confirms a strong effect of *tsc1a* knockdown so while there may be residual tonic levels of *Tsc1a* active

throughout the embryo we. Figure 11 summarises our findings which point toward distinct *tsc1a* and *ccr2* mediated mechanisms of susceptibility that have overlapping measurable phenotypes (bacterial burden, cell death, *tnfa*-expressing cells).

Cellular Heterogeneity:

Although *Tsc1* targeting reduces some of the effects of miR-126 inhibition, like bacterial survival and cell death, it does not inhibit macrophage recruitment of CCR2+ cells. Is this cell death observed in the *Tsc1* knockdown animal occurring in a different cellular population than the CCR2+ recruited cells, which are recruited in response to the cell death? And is it likely then that miR-126 controls expression of different targets in both macrophage/cellular populations? *TSCL1* in the "resident" population and a separate target linked to chemokine signalling in the CCR2+ cells?

This is a very interesting idea, however the zebrafish is not well suited to answer this particular question as there are (to our knowledge) no tools to track CCR2+ zebrafish cells. The best available technologies (RNAscope) are post mortem techniques that do not allow dynamic tracking and differentiation of responsive/resident cell populations. Our new data in Figure 8 showing independence of *tsc1a*/mTOR activity (measured as protection from cell death) from the *ccr2*-mediated migration phenotype (and any macrophage migration phenotype in the control animals) suggest the simple zebrafish embryo macrophage population may be fairly homogenous in this regard.

Direct miR-126 Targets:

This links to my final major point. The authors use bioinformatic analysis to predict miR-126 targets involved in the host response to mycobacterial infection in the zebrafish model. An initial mRNA screen validates some of these. *Tsc1* emerges as a strong candidate and an RNA-Hybrid plot is provided to illustrate the potential interaction. 3'UTR Luciferase Assays should be employed to confirm this, in a reporter cell system. Additionally, it is unclear from the manuscript if *CXCL12a* or *CCL2/CCR2* are thought to be direct targets of miR126 or their expression is regulated by other targets (the *Tsc1*/mTOR axis?). 3'UTR Luciferase assays would strengthen this and may point to the most relevant targets for miR-126.

Both *Tsc1* and *Cxcl12a* have previously been identified as direct targets of miR-126. A study in zebrafish has shown direct targeting of *Cxcl12a* by miR-126a (doi: 10.1161/ATVBAHA.116.308120) while multiple studies have identified *Tsc1* as a target of miR-126 in mice, humans, and pigs (doi: 10.1038/ni.2767) (doi: 10.1007/s11626-018-0292-0). As the degree of sequence homology and apparent conservation of target genes is high, we believe that *Tsc1* is a direct target of miR-126 in zebrafish. This has been further mentioned in the text.

We made two attempts at RNAseq analysis of miR-126 knockdown embryos which generated only very truncated lists of DEGs. Surprisingly despite using RNA samples (generated from two countries) that showed differential expression of *tsc1a*, *cxcr4b*, and *cxcl12a* by our conventional cDNA synthesis/qPCR measurement, these transcripts were all found to be present at equivalent levels by RNAseq measurement.

Furthermore, we were unable to perform experiments showing direct binding of zebrafish miR-126 to anything in cell culture transfection experiments using the predicted target sequence as bait in dual luciferase assays.

In light of these failures we have rewritten the initial sections to include mention of the uncertainty of a miR-126/*tsc1a* axis and to refocus the writing of the overall manuscript to focus on the downstream *tsc1a* and *cxcl12a* phenotypes.

Minor points for correction/clarification:

Intro; line 58/59; would it be better to describe microRNAs as "fine-tuners" rather than "master regulators"

Changed to "fine-tuners"

Line 72/73; any information on which cells in cattle decreased miR-126 was observed in
Line 75/76; similarly, what cells or tissues in humans?

miR-126 was decreased in serum from cattle, while human and mouse studies used either serum/plasma/CSF or PBMC/THP-1 macrophages to assess miRNA expression. miR-126 was believed to be specifically expressed in endothelial cells, however it is likely that while much of the miR-126 is from these endothelial cells, during infection or exposure to pathogens, it is also produced by immune cells such as dendritic cells and macrophages.

Results, Fig 7c: in the recruited macrophage reporter mouse, the trace does not appear to show increased recruitment in control (non-knockdown) mice, is this expected?

Yes, our data shows the timepoints we looked at have a plateau of macrophage recruitment to nascent granulomas in the control animals. This is quite interesting to think about in hindsight as we genuinely did not notice it. A quick literature review shows this is to be expected with relatively little change in macrophage migration over a 2 hour period reported by Hu et al *Frontiers in Immunology* 2023, and a 4 hour period shown in Saelens, Sweeney, and Viswanathan et al *Cell* 2022. Our videos demonstrate the process is as dynamic as expected for later growing granulomas but the overall number of macrophages appears to be balanced during this earliest stage of granuloma formation.

December 13, 2023

Re: Life Science Alliance manuscript #LSA-2023-02523-T

Dr. Stefan H Oehlers
Agency for Science, Technology and Research
A*STAR Infectious Diseases Labs
8A Biomedical Grove, Immunos Building
Level 5
Singapore, NSW 138648

Dear Dr. Oehlers,

Thank you for submitting your manuscript entitled "miR-126 suppresses tsc1- and cxcl12a-dependent susceptibility to mycobacterial infection" to Life Science Alliance. We invite you to submit a revised manuscript addressing the following Reviewer comments:

- Address Reviewer 1's comments.
- Address Reviewer 2's Major concern #4, and the Minor concerns.

Thank you for this interesting contribution to Life Science Alliance. We are looking forward to receiving your revised manuscript.

Sincerely,

- A letter addressing the reviewers' comments point by point.
- An editable version of the final text (.DOC or .DOCX) is needed for copyediting (no PDFs).
- High-resolution figure, supplementary figure and video files uploaded as individual files: See our detailed guidelines for preparing your production-ready images, <https://www.life-science-alliance.org/authors>
- Summary blurb (enter in submission system): A short text summarizing in a single sentence the study (max. 200 characters including spaces). This text is used in conjunction with the titles of papers, hence should be informative and complementary to the title and running title. It should describe the context and significance of the findings for a general readership; it should be written in the present tense and refer to the work in the third person. Author names should not be mentioned.
- By submitting a revision, you attest that you are aware of our payment policies found here: <https://www.life-science-alliance.org/copyright-license-fee>

B. MANUSCRIPT ORGANIZATION AND FORMATTING:

Response to reviewers

Referee #1:

The authors have address most, however not all of the concerns, some not even with arguments (which is at least surprising given the long revision time). We appreciate the attempts to improve the paper with RNA sequencing. Due to rather mysterious methodological difficulties, no results could be obtained for sequencing of miR-126 knockdown embryos and the manuscript was reoriented with a focus on the impact on *tsc1* and *cxcl12* instead. Several major issues remain, some of which were already pointed out in the first round of reviews.

1. Although the authors failed to prove the existence of a direct miR-126/*tsc1* axis and realigned the manuscript accordingly, this change in overall direction is not reflected by the current title. This needs to be changes

Title has been changed to “Zebrafish *tsc1* and *cxcl12a* increase susceptibility to mycobacterial infection”.

2. The apparently uninterpretable RNA seq analysis remains a mystery ("very truncated lists of DEGs"). Did the authors fail to extract RNA in sufficient quantity/quality? If so why? What was exactly tried (e.g. scaling up)? Which DEG were identified? This needs to be clarified, i.e. data need to be presented in the supplement.

We gone back to our initial RNAseq dataset from 2021 to create Supplementary Figure 2 to cover this dataset because if gives the most DEGs. Raw sequencing information has been deposited in GEO. Subsequent RNAseq replicates of the same conditions yielded fewer DEGs and aggregation alongside this initial run further depleted the list of DEGs. We attribute this diminishing returns to expiry of knockdown and molecular biology reagents after the primary lab group closed.

We note that our initial RNAseq dataset confirm the known roles for miR-126 in controlling neurobiology and Notch signalling (haematopoiesis and blood vessel formation).

3. The reviewer asked for the original expression data in uninfected embryos (e.g. relative expression). These data are still missing. Therefore, the data are requested again as supplementary information together with an explanation on how the fold change for uninfected controls was calculated (e.g. in figure 2A-E).

Original request: “It seems essential to see the original expression data in uninfected embryos, i.e. relative expression in uninfected controls related to house-keeping genes as data (e.g. Fig. 1A + B).”

New text added to materials and methods, we think this clarifies the relative quantification calculation method and conversion to fold change:

U6 or β -actin was used as an endogenous control for normalisation and data analysed using the $2^{-\Delta\Delta}$ Ct method with an average of controls used to set baseline and fold change was calculated as the ratio of the difference divided by the original value.

4. The reviewer asked for a further characterization of the macrophages (e.g. by iNOS). Any reply from the authors regarding this aspect is still missing.

We have added new text to the discussion “We did not examine other markers of macrophage polarisation such as *arg2* and *nos2a* which would allow further characterisation of the *tnfa:gfp* negative macrophage population in the miR-126 knockdown embryos” and added relevant

references to Irg1 (Sanderson LE et al, 2015) Arg2 (Hammond FR et al, 2023) and Nos2a (Elks PM et al, 2013) as markers of leukocyte polarisation in zebrafish.

5. The quality of the graphical abstract (figure 11) is not suitable for this journal. Please provide a complete, visually appealing graphical abstract.

Ouch, this one felt personal. Graphical abstracts are optional at LSA so it has been removed.

Additional minor aspects:

Line 45 Please correct: mTOR inhibition. We find

Corrected

Figure 1 B The legend is missing.

We have doubled checked the manuscript document and it seems to be there:

“Figure 1. Infection-induced miR-126 expression alters bacterial burden.

(A) Expression of miR-126 following *M. marinum* infection analysed by qPCR at 1 and 3 dpi relative to uninfected embryos.

(B) Expression of miR-126 in uninfected and infected, antagomir-injected (miR-126 knockdown) and scramble-injected embryos.

(C) Representative images of *M. marinum* infection at 3 dpi in scramble control and miR-126 knockdown embryos. Scale bar represents 200 μm .”

Figure 1 E Please remove statistics where $n = 2$; this is not suitable to perform statistical analysis. Should not Scram inf show around a two-fold increase compared to Scram uninf (compare to 1A)?

Statistical comparisons are removed. The datasets are totally different experiments/biological replicates so there are different scales of increases observed.

Line 373:wq Please correct: Our data thus far indicates

Corrected

Referee #2:

This manuscript demonstrated that miR-126 upregulation protects the zebrafish embryonic mycobacterium model. Two possible miR-126 targets Tsc1 and Cxcl12, mediate mTOR signaling or macrophage recruitment and activation, respectively. Suppression of either one in miR-126 deficient condition resulted in reduced bacteria burden. Although the importance of mTOR and CCR2 signaling in Mm infection in zebrafish models has been previously reported, the direct connection of these two pathways to miR-126 and its possible target tsc1 and cxcl12 is novel. However, the current study's significant weaknesses still need to be addressed.

Major concerns:

4, Figure 2 is problematic. tsc1a is the only one that showed significant upregulation upon miR-126 suppression. The remaining genes did not show significant changes upon Mm infection and miR-126 deletion. I suggest removing the non-significant data.

The *cxc* genes have upregulation upon miR-126 suppression in at least one of the conditions each. We have removed *spred1* as the only non-significant gene.

Minor concerns:

1. Figure 1E. Why would miR-126 sgRNA injection reduce its mRNA level?

We have added new text to the results. “We next utilised CRISPR-Cas9 to validate the effects of antagomir injection. Injection with guide RNAs targeting the miR-126 locus and Cas9 reduced the amount of miR-126 detectable by RT-qPCR indicating a possible combination of primer site mutation and transcript decay compared to embryos injected with scrambled guide RNAs and Cas 9 (Figure 1E).”

2. Figure 4 should include the transcript names and group them in functional groups in A and C to make the data useful.

Full DEG lists from Figures 4 A and C datasets are now provided in Supplementary File 2 along with GSEA tables as requested.

3. Data of miR-126 target prediction should be better organized using, for example, a Venn diagram to show the overlap of the predicted and validated targets rather than displaying the raw search results.

Supplementary File 1 has been modified with a new column in the “validated targets” sheet titled “Differential expression in this study” to highlight which transcripts were differentially expressed in miR-126 depleted animals in our study.

January 22, 2024

RE: Life Science Alliance Manuscript #LSA-2023-02523-TR

Dr. Stefan H Oehlers
Agency for Science, Technology and Research
A*STAR Infectious Diseases Labs
8A Biomedical Grove, Immunos Building
Level 5
Singapore 138648

Dear Dr. Oehlers,

Thank you for submitting your revised manuscript entitled "Zebrafish tsc1 and cxcl12a increase susceptibility to mycobacterial infection". We would be happy to publish your paper in Life Science Alliance pending final revisions necessary to meet our formatting guidelines.

- please be sure that the authorship listing and order is correct
- please be sure to include the References section
- please use the [10 author names et al.] format in your references (i.e., limit the author names to the first 10)
- please either upload figure 11, or remove its legend and any callouts to it in the text
- we encourage you to revise the figure legend for Figure 2 such that the figure panels are introduced in alphabetical order
- please add a callout for figures 2I; S1A-B; S2A-D; S3A-B and Table 1 to your main manuscript text

A. FINAL FILES:

B. MANUSCRIPT ORGANIZATION AND FORMATTING:

Sincerely,

January 23, 2024

RE: Life Science Alliance Manuscript #LSA-2023-02523-TRR

Dr. Stefan H Oehlers
Agency for Science, Technology and Research
A*STAR Infectious Diseases Labs
8A Biomedical Grove, Immunos Building
Level 5
Singapore 138648

Dear Dr. Oehlers,

Thank you for submitting your Research Article entitled "Zebrafish tsc1 and cxcl12a increase susceptibility to mycobacterial infection". It is a pleasure to let you know that your manuscript is now accepted for publication in Life Science Alliance. Congratulations on this interesting work.

DISTRIBUTION OF MATERIALS:

Again, congratulations on a very nice paper. I hope you found the review process to be constructive and are pleased with how the manuscript was handled editorially. We look forward to future exciting submissions from your lab.

Sincerely,
